# THE BREAK-EVEN POINT ON OPTIMIZATION TRAJECTORIES OF DEEP NEURAL NETWORKS

**Stanisław Jastrzębski**[1] , **Maciej Szymczak**[2], **Stanislav Fort**[3], **Devansh Arpit**[4], **Jacek Tabor**[2],
**Kyunghyun Cho**[1,5,6*], **Krzysztof Geras**[1*]
[1]New York University, USA
[2]Jagiellonian University, Poland
[3]Stanford University, USA
[4]Salesforce Research, USA
[5]Facebook AI Research, USA
[6]CIFAR Azrieli Global Scholar

## ABSTRACT

The early phase of training of deep neural networks is critical for their final performance. In this work, we study how the hyperparameters of stochastic gradient descent (SGD) used in the early phase of training affect the rest of the optimization trajectory. We argue for the existence of the "break-even" point on this trajectory, beyond which the curvature of the loss surface and noise in the gradient are implicitly regularized by SGD. In particular, we demonstrate on multiple classification tasks that using a large learning rate in the initial phase of training reduces the variance of the gradient, and improves the conditioning of the covariance of gradients. These effects are beneficial from the optimization perspective and become visible after the break-even point. Complementing prior work, we also show that using a low learning rate results in bad conditioning of the loss surface even for a neural network with batch normalization layers. In short, our work shows that key properties of the loss surface are strongly influenced by SGD in the early phase of training. We argue that studying the impact of the identified effects on generalization is a promising future direction.

## 1 INTRODUCTION

The connection between optimization and generalization of deep neural networks (DNNs) is not fully understood. For instance, using a large initial learning rate often improves generalization, which can come at the expense of the initial training loss reduction (Goodfellow et al., 2016; Li et al., 2019; Jiang et al., 2020). In contrast, using batch normalization layers typically improves both generalization and convergence speed of deep neural networks (Luo et al., 2019; Bjorck et al., 2018). These simple examples illustrate limitations of our understanding of DNNs.

Understanding the early phase of training has recently emerged as a promising avenue for studying the link between optimization and generalization of DNNs. It has been observed that applying regularization in the early phase of training is necessary to arrive at a well generalizing final solution (Keskar et al., 2017; Sagun et al., 2017; Achille et al., 2017). Another observed phenomenon is that the local shape of the loss surface changes rapidly in the beginning of training (LeCun et al., 2012; Keskar et al., 2017; Achille et al., 2017; Jastrzebski et al., 2018; Fort & Ganguli, 2019). Theoretical approaches to understanding deep networks also increasingly focus on the early part of the optimization trajectory (Li et al., 2019; Arora et al., 2019).

In this work, we study the dependence of the entire optimization trajectory on the early phase of training. We investigate noise in the mini-batch gradients using the covariance of gradients,[1] and the local curvature of the loss surface using the Hessian. These two matrices capture important and

---

*Equal contribution.

[1]We define it as $\mathbf{K} = \frac{1}{N} \sum_{i=1}^{N} (g_i - g)^T (g_i - g)$, where $g_i = g(\mathbf{x_i}, y_i; \theta)$ is the gradient of the training loss $\mathcal{L}$ with respect to $\theta$ on $x_i$, $N$ is the number of training examples, and $g$ is the full-batch gradient.

complementary aspects of optimization (Roux et al., 2008; Ghorbani et al., 2019) and generalization performance of DNNs (Jiang et al., 2020; Keskar et al., 2017; Bjorck et al., 2018; Fort et al., 2019). We include a more detailed discussion in Sec. 2.

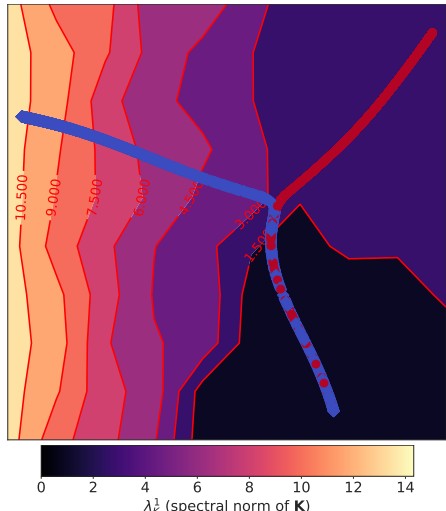 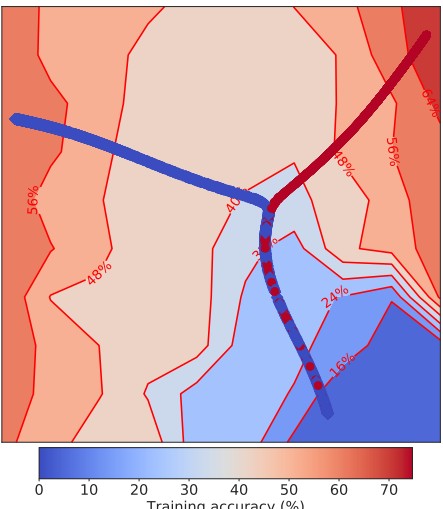

Figure 1: Visualization of the early part of the training trajectories on CIFAR-10 (before reaching 65% training accuracy) of a simple CNN model optimized using SGD with learning rates $\eta = 0.01$ (red) and $\eta = 0.001$ (blue). Each model on the training trajectory, shown as a point, is represented by its test predictions embedded into a two-dimensional space using UMAP. The background color indicates the spectral norm of the covariance of gradients $\mathbf{K}$ ($\lambda_K^1$, left) and the training accuracy (right). For lower $\eta$, after reaching what we call the break-even point, the trajectory is steered towards a region characterized by larger $\lambda_K^1$ (left) for the same training accuracy (right). See Sec. 4.1 for details. We also include an analogous figure for other quantities that we study in App. A.

Our first contribution is a simplified model of the early part of the training trajectory of DNNs. Based on prior empirical work (Sagun et al., 2017), we assume that the local curvature of the loss surface (the spectral norm of the Hessian) increases or decreases monotonically along the optimization trajectory. Under this model, gradient descent reaches a point in the early phase of training at which it oscillates along the most curved direction of the loss surface. We call this point the *break-even* point and show empirical evidence of its existence in the training of actual DNNs.

Our main contribution is to state and present empirical evidence for two conjectures about the dependence of the entire optimization trajectory on the early phase of training. Specifically, we conjecture that the hyperparameters of stochastic gradient descent (SGD) used before reaching the break-even point control: (1) the spectral norms of $\mathbf{K}$ and $\mathbf{H}$, and (2) the conditioning of $\mathbf{K}$ and $\mathbf{H}$. In particular, using a larger learning rate prior to reaching the break-even point reduces the spectral norm of $\mathbf{K}$ along the optimization trajectory (see Fig. 1 for an illustration of this phenomenon). Reducing the spectral norm of $\mathbf{K}$ decreases the variance of the mini-batch gradient, which has been linked to improved convergence speed (Johnson & Zhang, 2013).

Finally, we apply our analysis to a network with batch normalization (BN) layers and find that our predictions are valid in this case as well. Delving deeper in this line of investigation, we show that using a large learning rate is necessary to reach better-conditioned (relatively to a network without BN layers) regions of the loss surface, which was previously attributed to BN alone (Bjorck et al., 2018; Ghorbani et al., 2019; Page, 2019).

## 2 RELATED WORK

**Implicit regularization induced by the optimization method.** The choice of the optimization method implicitly affects generalization performance of deep neural networks (Neyshabur, 2017). In particular, using a large initial learning rate is known to improve generalization (Goodfellow et al.,

2016; Li et al., 2019). A classical approach to study these questions is to bound the generalization error using measures such as the norm of the parameters at the final minimum (Bartlett et al., 2017; Jiang et al., 2020).

An emerging approach is to study the properties of the whole optimization trajectory. Arora et al. (2019) suggest it is necessary to study the optimization trajectory to understand optimization and generalization of deep networks. In a related work, Erhan et al. (2010); Achille et al. (2017) show the existence of a critical period of learning. Erhan et al. (2010) argue that training, unless pretraining is used, is sensitive to shuffling of examples in the first epochs of training. Achille et al. (2017); Golatkar et al. (2019); Sagun et al. (2017); Keskar et al. (2017) demonstrate that adding regularization in the beginning of training affects the final generalization disproportionately more compared to doing so later. We continue research in this direction and study how the choice of hyperparameters in SGD in the early phase of training affects the optimization trajectory in terms of the covariance of gradients, and the Hessian.

**The covariance of gradients and the Hessian.** The Hessian quantifies the local curvature of the loss surface. Recent work has shown that the largest eigenvalues of $\mathbf{H}$ can grow quickly in the early phase of training (Keskar et al., 2017; Sagun et al., 2017; Fort & Scherlis, 2019; Jastrzebski et al., 2018). Keskar et al. (2017); Jastrzebski et al. (2017) studied the dependence of the Hessian (at the final minimum) on the optimization hyperparameters. The Hessian can be decomposed into two terms, where the dominant term (at least at the end of training) is the uncentered covariance of gradients $\mathbf{G}$ (Sagun et al., 2017; Papyan, 2019).

The covariance of gradients, which we denote by $\mathbf{K}$, encapsulates the geometry and the magnitude of variation in gradients across different samples. The matrix $\mathbf{K}$ was related to the generalization error in Roux et al. (2008); Jiang et al. (2020). Closely related quantities, such as the cosine alignment between gradients computed on different examples, were recently shown to explain some aspects of deep networks generalization (Fort et al., 2019; Liu et al., 2020; He & Su, 2020). Zhang et al. (2019) argues that in DNNs the Hessian and the covariance of gradients are close in terms of the largest eigenvalues.

**Learning dynamics of deep neural networks.** Our theoretical model is motivated by recent work on learning dynamics of neural networks (Goodfellow et al., 2014; Masters & Luschi, 2018; Wu et al., 2018; Yao et al., 2018; Xing et al., 2018; Jastrzebski et al., 2018; Lan et al., 2019). We are directly inspired by Xing et al. (2018) who show that for popular classification benchmarks, the cosine of the angle between consecutive optimization steps in SGD is negative. Similar observations can be found in Lan et al. (2019). Our theoretical analysis is inspired by Wu et al. (2018) who study how SGD selects the final minimum from a stability perspective. We apply their methodology to the early phase of training, and make predictions about the entire training trajectory.

## 3 THE BREAK-EVEN POINT AND THE TWO CONJECTURES ABOUT SGD TRAJECTORY

Our overall motivation is to better understand the connection between optimization and generalization of DNNs. In this section we study how the covariance of gradients ($\mathbf{K}$) and the Hessian ($\mathbf{H}$) depend on the early phase of training. We are inspired by recent empirical observations showing their importance for optimization and generalization of DNNs (see Sec. 2 for a detailed discussion).

Recent work has shown that in the early phase of training the gradient norm (Goodfellow et al., 2016; Fort & Ganguli, 2019; Liu et al., 2020) and the local curvature of the loss surface (Jastrzebski et al., 2018; Fort & Ganguli, 2019) can rapidly increase. Informally speaking, one scenario we study here is when this initial growth is rapid enough to destabilize training. Inspired by Wu et al. (2018), we formalize this intuition using concepts from dynamical stability. Based on the developed analysis, we state two conjectures about the dependence of $\mathbf{K}$ and $\mathbf{H}$ on hyperparameters of SGD, which we investigate empirically in Sec. 4.

**Definitions.** We begin by introducing the notation. Let us denote the loss on an example $(\mathbf{x}, y)$ by $\mathcal{L}(\mathbf{x}, y; \theta)$, where $\theta$ is a $D$-dimensional parameter vector. The two key objects we study are the Hessian of the training loss ($\mathbf{H}$), and the covariance of gradients $\mathbf{K} = \frac{1}{N} \sum_{i=1}^{N} (g_i - g)^T (g_i - g)$,

where $g_i = g(\mathbf{x_i}, y_i; \theta)$ is the gradient of $\mathcal{L}$ with respect to $\theta$ calculated on $i$-th example, $N$ is the number of training examples, and $g$ is the full-batch gradient. We denote the $i$-th normalized eigenvector and eigenvalue of a matrix $\mathbf{A}$ by $e_A^i$ and $\lambda_A^i$. Both $\mathbf{H}$ and $\mathbf{K}$ are computed at a given $\theta$, but we omit this dependence in the notation. Let $t$ index steps of optimization, and let $\theta(t)$ denote the parameter vector at optimization step $t$.

Inspired by Wu et al. (2018) we introduce the following condition to quantify *stability* at a given $\theta(t)$. Let us denote the projection of parameters $\theta$ onto $e_H^1$ by $\psi = \langle \theta, e_H^1 \rangle$. With a slight abuse of notation let $g(\psi) = \langle g(\theta), e_H^1 \rangle$. We say SGD is *unstable along* $e_H^1$ at $\theta(t)$ if the norm of elements of sequence $\psi(\tau + 1) = \psi(\tau) - \eta g(\psi(\tau))$ diverges when $\tau \to \infty$, where $\psi(0) = \theta(t)$. The sequence $\psi(\tau)$ represents optimization trajectory in which every step $t' > t$ is projected onto $e_H^1$.

**Assumptions.**  Based on recent empirical studies, we make the following assumptions.

1. The loss surface projected onto $e_H^1$ is a quadratic one-dimensional function of the form $f(\psi) = \sum_{i=1}^N (\psi - \psi^*)^2 H_i$. The same assumption was made in Wu et al. (2018), but for all directions in the weight space. Alain et al. (2019) show empirically that the loss averaged over all training examples is well approximated by a quadratic function along $e_H^1$.

2. The eigenvectors $e_H^1$ and $e_K^1$ are co-linear, i.e. $e_H^1 = \pm e_K^1$, and $\lambda_K^1 = \alpha \lambda_H^1$ for some $\alpha \in \mathbb{R}$. This is inspired by the fact that the top eigenvalues of $\mathbf{H}$ can be well approximated using $\mathbf{G}$ (non-centered $\mathbf{K}$) (Papyan, 2019; Sagun et al., 2017). Zhang et al. (2019) shows empirical evidence for co-linearity of the largest eigenvalues of $\mathbf{K}$ and $\mathbf{H}$.

3. If optimization is not stable along $e_H^1$ at a given $\theta(t)$, $\lambda_H^1$ decreases in the next step, and the distance to the minimum along $e_H^1$ increases in the next step. This is inspired by recent work showing training can escape a region with too large curvature compared to the learning rate (Zhu et al., 2018; Wu et al., 2018; Jastrzebski et al., 2018).

4. The spectral norm of $\mathbf{H}$, $\lambda_H^1$, increases during training and the distance to the minimum along $e_H^1$ decreases, unless increasing $\lambda_H^1$ would lead to entering a region where training is not stable along $e_H^1$. This is inspired by (Keskar et al., 2017; Goodfellow et al., 2016; Sagun et al., 2017; Jastrzebski et al., 2018; Fort & Scherlis, 2019; Fort & Ganguli, 2019) who show that in many settings $\lambda_H^1$ or gradient norm increases in the beginning of training, while at the same time the overall training loss decreases.

Finally, we also assume that $S \gg N$, i.e. that the batch size is small compared to the number of training examples. These assumptions are only used to build a theoretical model for the early phase of training. Its main purpose is to make predictions about the training procedure that we test empirically in Sec. 4.

**Reaching the break-even point earlier for a larger learning rate or a smaller batch size.**  Let us restrict ourselves to the case when training is initialized at $\theta(0)$ at which SGD is stable along $e_H^1(0)$.[2] We aim to show that the learning rate ($\eta$) and the batch size ($S$) determine $\mathbf{H}$ and $\mathbf{K}$ in our model, and conjecture that the same holds empirically for realistic neural networks.

Consider two optimization trajectories for $\eta_1$ and $\eta_2$, where $\eta_1 > \eta_2$, that are initialized at the same $\theta_0$, where optimization is stable along $e_H^1(t)$ and $\lambda_H^1(t) > 0$. Under Assumption 1 the loss surface along $e_H^1(t)$ can be expressed as $f(\psi) = \sum_{i=1}^N (\psi - \psi^*)^2 H_i(t)$, where $H_i(t) \in \mathbb{R}$. It can be shown that at any iteration $t$ the necessary and sufficient condition for SGD to be stable along $e_H^1(t)$ is:

$$(1 - \eta \lambda_H^1(t))^2 + s(t)^2 \frac{\eta^2(N - S)}{S(N - 1)} \leq 1, \tag{1}$$

where $N$ is the training set size and $s(t)^2 = \mathrm{Var}[H_i(t)]$ over the training examples. A proof can be found in (Wu et al., 2018). We call this point on the trajectory on which the LHS of Eq. 1 becomes equal to 1 for the first time the *break-even point*. By definition, there exists only a single break-even point on the training trajectory.

Under Assumption 3, $\lambda_H^1(t)$ and $\lambda_K^1(t)$ increase over time. If $S = N$, the break-even point is reached at $\lambda_H^1(t) = \frac{2}{\eta}$. More generally, it can be shown that for $\eta_1$, the break-even point is reached

---

[2]We include a similar argument for the opposite case in App. B.

for a lower magnitude of $\lambda_H^1(t)$ than for $\eta_2$. The same reasoning can be repeated for $S$ (in which case we assume $N \gg S$). We state this formally and prove in App. B.

Under Assumption 4, after passing the break-even point on the training trajectory, SGD does not enter regions where either $\lambda_H^1$ or $\lambda_K^1$ is larger than at the break-even point, as otherwise it would lead to increasing one of the terms in LHS of Eq. 1, and hence losing stability along $e_H^1$.

**Two conjectures about real DNNs.** Assuming that real DNNs reach the break-even point, we make the following two conjectures about their optimization trajectory.

The most direct implication of reaching the break-even point is that $\lambda_K^1$ and $\lambda_H^1$ at the break-even point depend on $\eta$ and $S$, which we formalize as:

**Conjecture 1** (Variance reduction effect of SGD). *Along the SGD trajectory, the maximum attained values of $\lambda_H^1$ and $\lambda_K^1$ are smaller for a larger learning rate or a smaller batch size.*

We refer to Conjecture 1 as *variance reduction effect of SGD* because reducing $\lambda_K^1$ can be shown to reduce the $L_2$ distance between the full-batch gradient, and the mini-batch gradient. We expect that similar effects exist for other optimization or regularization methods. We leave investigating them for future work.

Next, we make another, stronger, conjecture. It is plausible to assume that reaching the break-even point affects to a lesser degree $\lambda_H^i$ and $\lambda_K^i$ for $i \neq 1$ because increasing their values does not impact stability along $e_H^1$. Based on this we conjecture that:

**Conjecture 2** (Pre-conditioning effect of SGD). *Along the SGD trajectory, the maximum attained values of $\frac{\lambda_K^*}{\lambda_K^1}$ and $\frac{\lambda_H^*}{\lambda_H^1}$ are larger for a larger learning rate or a smaller batch size, where $\lambda_K{}^*$ and $\lambda_H{}^*$ are the smallest non-zero eigenvalues of $\mathbf{K}$ and $\mathbf{H}$, respectively. Furthermore, the maximum attained values of $\mathrm{Tr}(\mathbf{K})$ and $\mathrm{Tr}(\mathbf{H})$ are smaller for a larger learning rate or a smaller batch size.*

We consider non-zero eigenvalues in the conjecture, because $\mathbf{K}$ has at most $N - 1$ non-zero eigenvalues, where $N$ is the number of training points, which can be much smaller than $D$ in overparametrized DNNs. Both conjectures are valid only for learning rates and batch sizes that guarantee that training converges.

From the optimization perspective, the effects discussed above are desirable. Many papers in the optimization literature underline the importance of reducing the variance of the mini-batch gradient (Johnson & Zhang, 2013) and the conditioning of the covariance of gradients (Roux et al., 2008). There also exists a connection between these effects and generalization (Jiang et al., 2020), which we discuss towards the end of the paper.

## 4 EXPERIMENTS

In this section we first analyse learning dynamics in the early phase of training. Next, we empirically investigate the two conjectures. In the final part we extend our analysis to a neural network with batch normalization layers.

We run experiments on the following datasets: CIFAR-10 (Krizhevsky, 2009), IMDB dataset (Maas et al., 2011), ImageNet (Deng et al., 2009), and MNLI (Williams et al., 2018). We apply to these datasets the following architectures: a vanilla CNN (SimpleCNN) following Keras example (Chollet et al., 2015), ResNet-32 (He et al., 2015), LSTM (Hochreiter & Schmidhuber, 1997), DenseNet (Huang et al., 2016), and BERT (Devlin et al., 2018). We also include experiments using a multi-layer perceptron trained on the FashionMNIST dataset (Xiao et al., 2017) in the Appendix. All experimental details are described in App. D.

Following Dauphin et al. (2014); Alain et al. (2019), we estimate the top eigenvalues and eigenvectors of $\mathbf{H}$ on a small subset of the training set (e.g. $5\%$ in the case of CIFAR-10) using the Lanczos algorithm (Lanczos, 1950). As computing the full eigenspace of $\mathbf{K}$ is infeasible for real DNNs, we compute the covariance using mini-batch gradients. In App. C we show empirically that (after normalization) this approximates well the largest eigenvalue, and we include other details on computing the eigenspaces.

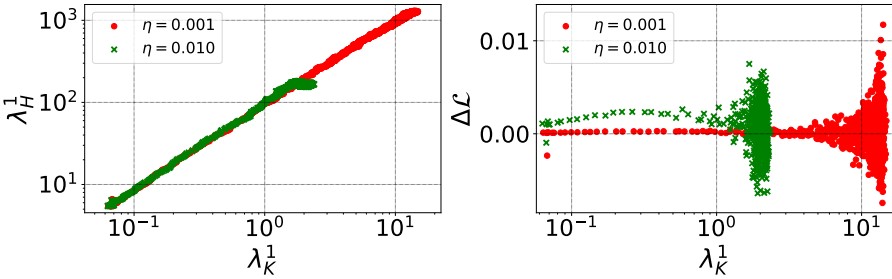

Figure 2: The spectral norm of $\mathbf{H}$ ($\lambda_H^1$, left) and $\Delta\mathcal{L}$ (difference in the training loss computed between two consecutive steps, right) versus $\lambda_K^1$ at different training iterations. Experiment was performed with SimpleCNN on the CIFAR-10 dataset with two different learning rates (color). Consistently with our theoretical model, $\lambda_K^1$ is correlated initially with $\lambda_H^1$, and training is generally stable ($\Delta\mathcal{L} > 0$) prior to achieving the maximum value of $\lambda_K^1$.

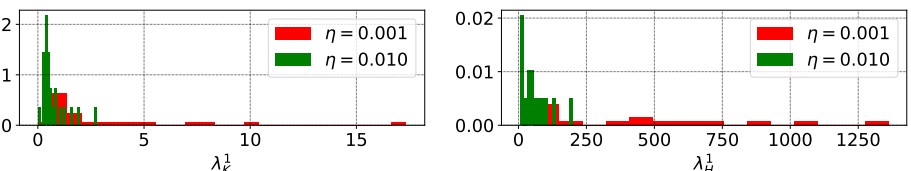

Figure 3: The spectrum of $\mathbf{K}$ (left) and $\mathbf{H}$ (right) at the training iteration corresponding to the largest value of $\lambda_K^1$ and $\lambda_H^1$, respectively. Experiment was performed with SimpleCNN on the CIFAR-10 dataset with two different learning rates (color). Consistently with Conjecture 2, training with lower learning rate results in finding a region of the loss surface characterized by worse conditioning of $\mathbf{K}$ and $\mathbf{H}$ (visible in terms of the large number of "spikes" in the spectrum, see also Fig. 4).

## 4.1 A CLOSER LOOK AT THE EARLY PHASE OF TRAINING

First, we examine the learning dynamics in the early phase of training. Our goal is to verify some of the assumptions made in Sec. 3. We analyse the evolution of $\lambda_H^1$ and $\lambda_K^1$ when using $\eta = 0.01$ and $\eta = 0.001$ to train SimpleCNN on the CIFAR-10 dataset. We repeat this experiment 5 times using different random initializations of network parameters.

**Visualizing the break-even point.** We visualize the early part of the optimization trajectory in Fig. 1. Following Erhan et al. (2010), we embed the test set predictions at each step of training of SimpleCNN using UMAP (McInnes et al., 2018). The background color indicates $\lambda_K^1$ (left) and the training accuracy (right) at the iteration with the closest embedding in Euclidean distance.

We observe that the trajectory corresponding to the lower learning rate reaches regions of the loss surface characterized by larger $\lambda_K^1$, compared to regions reached at the same training accuracy in the second trajectory. Additionally, in Fig. 3 we plot the spectrum of $\mathbf{K}$ (left) and $\mathbf{H}$ (right) at the iterations when $\lambda_K$ and $\lambda_H$ respectively reach the highest values. We observe more outliers for the lower learning rate in the distributions of both $\lambda_K$ and $\lambda_H$.

**Are $\lambda_K^1$ and $\lambda_H^1$ correlated in the beginning of training?** The key assumption behind our theoretical model is that $\lambda_K^1$ and $\lambda_H^1$ are correlated, at least prior to reaching the break-even point. We confirm this in Fig. 2. The highest achieved $\lambda_K^1$ and $\lambda_H^1$ are larger for the smaller $\eta$. Additionally, we observe that after achieving the highest value of $\lambda_H^1$, further growth of $\lambda_K^1$ does not translate to an increase of $\lambda_H^1$. This is expected as $\lambda_H^1$ decays to 0 when the mean loss decays to 0 for cross entropy loss (Martens, 2016).

**Does training become increasingly unstable in the early phase of training?** According to Assumption 3, an increase of $\lambda_K^1$ and $\lambda_H^1$ translates into a decrease in stability, which we formalized as stability along $e_H^1$. Computing stability along $e_H^1$ directly is computationally expensive. Instead, we measure a more tractable proxy. At each iteration we measure the loss on the training set before and

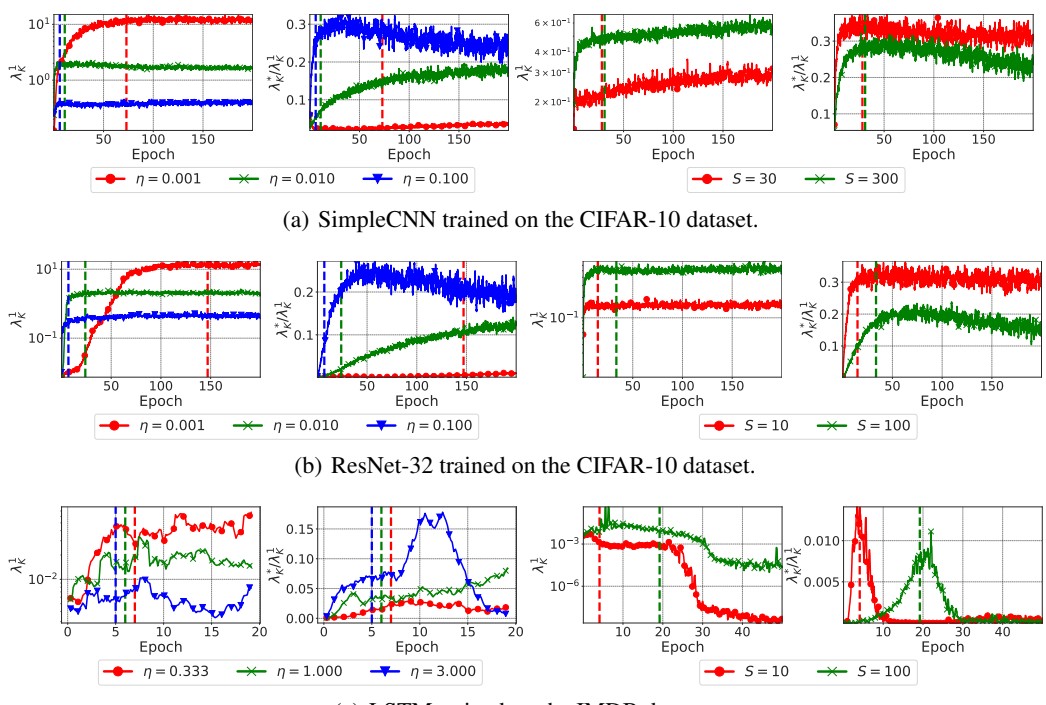

(a) SimpleCNN trained on the CIFAR-10 dataset.

(b) ResNet-32 trained on the CIFAR-10 dataset.

(c) LSTM trained on the IMDB dataset.

Figure 4: The *variance reduction* and the *pre-conditioning* effect of SGD in various settings. The optimization trajectories corresponding to higher learning rates ($\eta$) or lower batch sizes ($S$) are characterized by lower maximum $\lambda_K^1$ (the spectral norm of the covariance of gradients) and larger maximum $\lambda_K^*/\lambda_K^1$ (the condition number of the covariance of gradients). Vertical lines mark epochs at which the training accuracy is larger (for the first time) than a manually picked threshold, which illustrates that the effects are not explained by differences in training speeds.

after taking the step, which we denote as $\Delta\mathcal{L}$ (a positive value indicates a reduction of the training loss). In Fig. 2 we observe that training becomes increasingly unstable ($\Delta\mathcal{L}$ starts to take negative values) as $\lambda_K^1$ reaches the maximum value.

**Summary.** We have shown that the early phase of training is consistent with the assumptions made in our theoretical model. That is, $\lambda_K^1$ and $\lambda_H^1$ increase approximately proportionally to each other, which is also generally correlated with a decrease of a proxy of stability. Finally, we have shown qualitatively reaching the break-even point.

## 4.2 THE VARIANCE REDUCTION AND THE PRE-CONDITIONING EFFECT OF SGD

In this section we test empirically Conjecture 1 and Conjecture 2. For each model we manually pick a suitable range of learning rates and batch sizes to ensure that the properties of $\mathbf{K}$ and $\mathbf{H}$ that we study have converged under a reasonable computational budget. We mainly focus on studying the covariance of gradients ($\mathbf{K}$), and leave a closer investigation of the Hessian for future work. We use the batch size of $128$ to compute $\mathbf{K}$ when we vary the batch size for training. When we vary the learning rate instead, we use the same batch size as the one used to train the model. App. C describes the remaining details on how we approximate the eigenspaces of $\mathbf{K}$ and $\mathbf{H}$.

We summarize the results for SimpleCNN, ResNet-32, LSTM, BERT, and DenseNet in Fig. 4, Fig. 5, and Fig. 6. Curves are smoothed using moving average for clarity. Training curves and additional experiments are reported in App. E.

**Testing Conjecture 1.** To test Conjecture 1, we examine the highest value of $\lambda_K^1$ observed along the optimization trajectory. As visible in Fig. 4, using a higher $\eta$ results in $\lambda_K^1$ achieving a lower

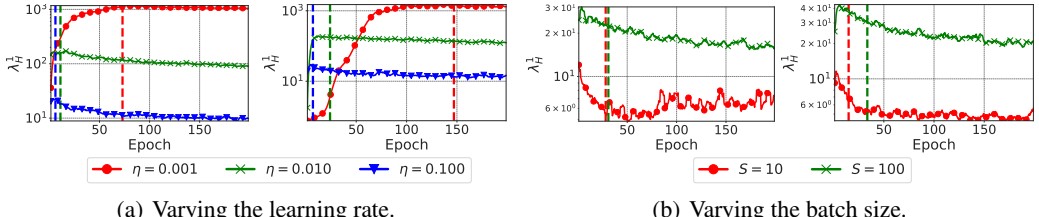

(a) Varying the learning rate.  (b) Varying the batch size.

Figure 5: The *variance reduction* effect of SGD, for ResNet-32 and SimpleCNN. Trajectories corresponding to higher learning rates ($\eta$, left) or smaller batch sizes ($S$, right) are characterized by a lower maximum $\lambda_H^1$ (the spectral norm of the Hessian) along the trajectory. Vertical lines mark epochs at which the training accuracy is larger (for the first time) than a manually picked threshold.

maximum during training. Similarly, we observe that using a higher $S$ in SGD leads to reaching a higher maximum value of $\lambda_K^1$. For instance, for SimpleCNN (top row of Fig. 4) we observe $\max(\lambda_K^1) = 0.68$ and $\max(\lambda_K^1) = 3.30$ for $\eta = 0.1$ and $\eta = 0.01$, respectively.

**Testing Conjecture 2.** To test Conjecture 2, we compute the maximum value of $\lambda_K^*/\lambda_K^1$ along the optimization trajectory. It is visible in Fig. 4 that using a higher $\eta$ results in reaching a larger maximum value of $\lambda_K^*/\lambda_K^1$ along the trajectory. For instance, in the case of SimpleCNN $\max(\lambda_K^*/\lambda_K^1) = 0.37$ and $\max(\lambda_K^*/\lambda_K^1) = 0.24$ for $\eta = 0.1$ and $\eta = 0.01$, respectively.

**A counter-intuitive effect of decreasing the batch size.** Consistently with Conjecture 2, we observe that the maximum value of $\mathrm{Tr}(\mathbf{K})$ is *smaller* for the smaller batch size. In the case of SimpleCNN $\max(\mathrm{Tr}(\mathbf{K})) = 5.56$ and $\max(\mathrm{Tr}(\mathbf{K})) = 10.86$ for $S = 10$ and $S = 100$, respectively. Due to space constraints we report the effect of $\eta$ and $S$ on $\mathrm{Tr}(\mathbf{K})$ in other settings in App. E.

This effect is counter-intuitive because $\mathrm{Tr}(\mathbf{K})$ is proportional to the variance of the mini-batch gradient (see also App. C). Naturally, using a lower batch size generally *increases* the variance of the mini-batch gradient, and $\mathrm{Tr}(\mathbf{K})$. This apparent contradiction is explained by the fact that we measure $\mathrm{Tr}(\mathbf{K})$ using a different batch size (128) than the one used to train the model. Hence, decreasing the batch size both increases (due to approximating the gradient using fewer samples) and decreases (as predicted in Conjecture 2) the variance of the mini-batch gradient along the optimization trajectory.

**How early in training is the break-even point reached?** We find that $\lambda_K^1$ and $\lambda_H^1$ reach their highest values early in training, close to reaching $60\%$ training accuracy on CIFAR-10, and $75\%$ training accuracy on IMDB. The training and validation accuracies are reported for all the experiments in App. E. This suggests that the break-even point is reached early in training.

**The Hessian.** In the above, we have focused on the covariance of gradients. In Fig. 5 we report how $\lambda_H^1$ depends on $\eta$ and $S$ for ResNet-32 and SimpleCNN. Consistently with prior work (Keskar et al., 2017; Jastrzebski et al., 2018), we observe that using a smaller $\eta$ or using a larger $S$ coincides with a larger maximum value of $\lambda_H^1$. For instance, for SimpleCNN we observe $\max(\lambda_H^1) = 26.27$ and $\max(\lambda_H^1) = 211.17$ for $\eta = 0.1$ and $\eta = 0.01$, respectively. We leave testing predictions made in Conjecture 2 about the Hessian for future work.

**Larger scale studies.** Finally, we test the two conjectures in two larger scale settings: BERT fine-tuned on the MNLI dataset, and DenseNet trained on the ImageNet dataset. Due to memory constraints, we only vary the learning rate. We report results in Fig. 6. We observe that both conjectures hold in these two settings. It is worth noting that DenseNet uses batch normalization layers. In the next section we investigate closer batch-normalized networks.

**Summary.** In this section we have shown evidence supporting the variance reduction (Conjecture 1) and the pre-conditioning effect (Conjecture 2) of SGD in a range of classification tasks. We also found that the above conclusions hold for MLP trained on the Fashion MNIST dataset, SGD with momentum, and SGD with learning rate decay. We include these results in App. E-G.

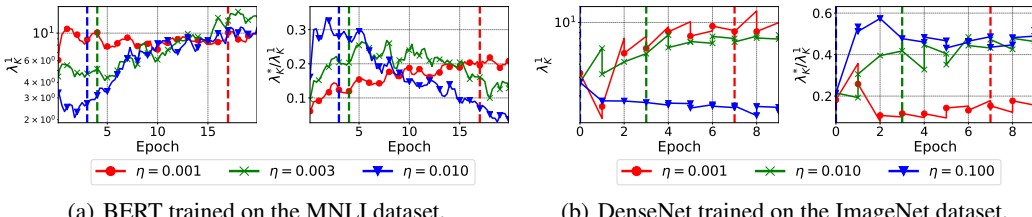

(a) BERT trained on the MNLI dataset.  (b) DenseNet trained on the ImageNet dataset.

Figure 6: The *variance reduction* and the *pre-conditioning* effect of SGD, demonstrated on two larger scale settings: BERT on the MNLI dataset (left), and DenseNet on the ImageNet dataset (right). For each setting we report $\lambda_K^1$ (left) and $\lambda_K^*/\lambda_K^1$ (right). Vertical lines mark epochs at which the training accuracy is larger (for the first time) than a manually picked threshold.

## 4.3 Importance of Learning Rate for Conditioning in Networks with Batch Normalization Layers

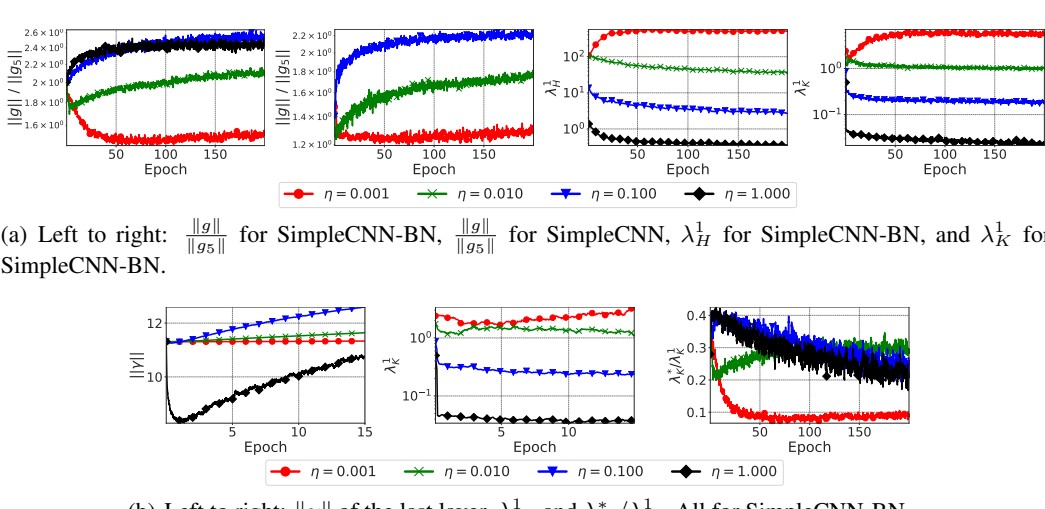

(a) Left to right: $\frac{\|g\|}{\|g_5\|}$ for SimpleCNN-BN, $\frac{\|g\|}{\|g_5\|}$ for SimpleCNN, $\lambda_H^1$ for SimpleCNN-BN, and $\lambda_K^1$ for SimpleCNN-BN.

(b) Left to right: $\|\gamma\|$ of the last layer, $\lambda_K^1$, and $\lambda_K^*/\lambda_K^1$. All for SimpleCNN-BN.

Figure 7: The effect of changing the learning rate on various metrics (see text for details) for SimpleCNN with and without batch normalization layers (SimpleCNN-BN and SimpleCNN).

The loss surface of deep neural networks has been widely reported to be ill-conditioned (LeCun et al., 2012; Martens, 2016). Recently, Ghorbani et al. (2019); Page (2019) argued that the key reason behind the efficacy of batch normalization (Ioffe & Szegedy, 2015) is improving conditioning of the loss surface. In Conjecture 2 we suggest that using a high $\eta$ (or a small $S$) results in improving the conditioning of $\mathbf{K}$ and $\mathbf{H}$. A natural question that we investigate in this section is how the two phenomena are related. We study here the effect of learning rate, and report in App. H an analogous study for batch size.

**Are the two conjectures valid in networks with batch normalization layers?** First, to investigate whether our conjectures hold in networks with batch normalization layers, we run similar experiments as in Sec. 4.2 with a SimpleCNN model with batch normalization layers inserted after each layer (SimpleCNN-BN), on the CIFAR-10 dataset. We test $\eta \in \{0.001, 0.01, 0.1, 1.0\}$ (using $\eta = 1.0$ leads to divergence of SimpleCNN without BN). We summarize the results in Fig. 7. We observe that the evolution of $\lambda_K^*/\lambda_K^1$ and $\lambda_K^1$ is consistent with both Conjecture 1 and Conjecture 2.

**A closer look at the early phase of training.** To further corroborate that our analysis applies to networks with batch normalization layers, we study the early phase of training of SimpleCNN-BN, complementing the results in Sec. 4.1.

We observe in Fig. 7 (bottom) that training of SimpleCNN-BN starts in a region characterized by a relatively high $\lambda_K^1$. This is consistent with prior work showing that networks with batch normalization layers can exhibit gradient explosion in the first iteration (Yang et al., 2019). The value of $\lambda_K^1$ then decays for all but the lowest $\eta$. This behavior is consistent with our theoretical model. We also track the norm of the scaling factor in the batch normalization layers, $\|\gamma\|$, in the last layer of the network in Fig. 7 (bottom). It is visible that $\eta = 1.0$ and $\eta = 0.1$ initially decrease the value of $\|\gamma\|$, which we hypothesize to be one of the mechanisms due to which high $\eta$ steers optimization towards better conditioned regions of the loss surface in batch-normalized networks. Interestingly, this seems consistent with Luo et al. (2019) who argue that using mini-batch statistics in batch normalization acts as an implicit regularizer by reducing $\|\gamma\|$.

**Using batch normalization requires using a high learning rate.** As our conjectures hold for SimpleCNN-BN, a natural question is if the loss surface can be ill-conditioned with a low learning rate even when batch normalization is used. Ghorbani et al. (2019) show that without batch normalization, mini-batch gradients are largely contained in the subspace spanned by the top eigenvectors of noncentered $\mathbf{K}$. To answer this question we track $\|g\|/\|g_5\|$, where $g$ denotes the mini-batch gradient, and $g_5$ denotes the mini-batch gradient projected onto the top 5 eigenvectors of $\mathbf{K}$. A value of $\|g\|/\|g_5\|$ close to 1 implies that the mini-batch gradient is mostly contained in the subspace spanned by the top 5 eigenvectors of $\mathbf{K}$.

We compare two settings: SimpleCNN-BN optimized with $\eta = 0.001$, and SimpleCNN optimized with $\eta = 0.01$. We make three observations. First, the maximum and minimum values of $\|g\|/\|g_5\|$ are 1.90 (1.37) and 2.02 (1.09), respectively. Second, the maximum and minimum values of $\lambda_K^1$ are 12.05 and 3.30, respectively. Finally, $\lambda_K^*/\lambda_K^1$ reaches 0.343 in the first setting, and 0.24 in the second setting. Comparing these differences to differences that are induced by using the highest $\eta = 1.0$ in SimpleCNN-BN, we can conclude that using a large learning rate is necessary to observe the effect of loss smoothing which was previously attributed to batch normalization alone (Ghorbani et al., 2019; Page, 2019; Bjorck et al., 2018). This might be directly related to the result that using a high learning rate is necessary to achieve good generalization when using batch normalization layers (Bjorck et al., 2018).

**Summary.** We have shown that the effects of the learning rate predicted in Conjecture 1 and Conjecture 2 hold for a network with batch normalization layers, and that *using a high learning rate is necessary in a network with batch normalization layers to improve conditioning of the loss surface, compared to conditioning of the loss surface in the same network without batch normalization layers.*

## 5 CONCLUSION

Based on our theoretical model, we argued for the existence of the break-even point on the optimization trajectory induced by SGD. We presented evidence that hyperparameters used in the early phase of training control the spectral norm and the conditioning of $\mathbf{K}$ (a matrix describing noise in the mini-batch gradients) and $\mathbf{H}$ (a matrix describing local curvature of the loss surface) after reaching the break-even point. In particular, using a large initial learning rate steers training to better conditioned regions of the loss surface, which is beneficial from the optimization point of view.

A natural direction for the future is connecting our observations to recent studies on the relation of measures, such as gradient variance, to the generalization of deep networks (Li et al., 2019; Jiang et al., 2020; Fort et al., 2019). Our work shows that the hyperparameters of SGD control these measures after the break-even point. Another interesting direction is to understand the connection between the existence of the break-even point and the existence of the *critical learning period* in training of DNNs (Achille et al., 2017).

ACKNOWLEDGMENTS

KC thanks NVIDIA and eBay for their support. SJ thanks Amos Storkey and Luke Darlow for fruitful discussions.

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

## A  ADDITIONAL VISUALIZATION OF THE BREAK-EVEN POINT

We include here an analogous Figure to Fig. 1, but visualizing the conditioning of the covariance of gradients ($\lambda_K^*/\lambda_K^1$, left), the trace of the covariance of gradients ($\text{Tr}(\mathbf{K})$, middle), and the spectral norm of the Hessian ($\lambda_H^1$, right).

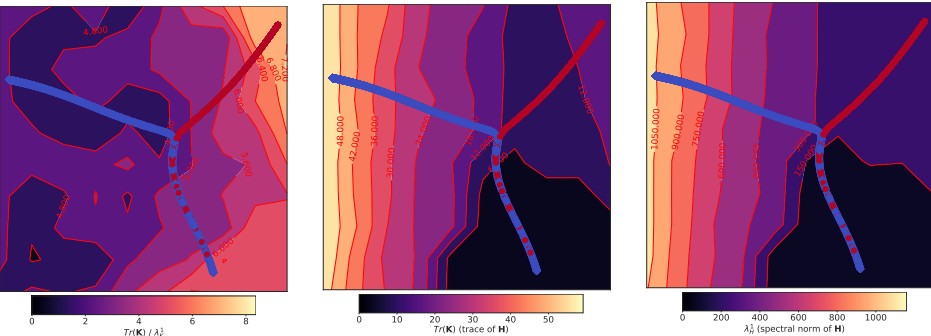

Figure 8: Analogous to Fig. 1. The background color indicates the conditioning of the covariance of gradients $\mathbf{K}$ ($\lambda_K^*/\lambda_K^1$, left), the trace of the covariance of gradients ($\text{Tr}(\mathbf{K})$, middle), and the spectral norm of the Hessian ($\lambda_H^1$, right).

## B  PROOFS

In this section we formally state and prove the theorem used in Sec. 3. With the definitions introduced in Sec. 3 in mind, we state the following:

**Theorem 1.** *Assuming that training is stable along $e_H^1(t)$ at $t = 0$, then $\lambda_H^1(t)$ and $\lambda_K^1(t)$ at which SGD becomes unstable along $e_H^1(t)$ are smaller for a larger $\eta$ or a smaller $S$.*

*Proof.* This theorem is a straightforward application of Theorem 1 from Wu et al. (2018) to the early phase of training. First, let us consider two optimization trajectories corresponding to using two different learning rates $\eta_1$ and $\eta_2$ ($\eta_1 > \eta_2$). For both trajectories, Theorem 1 of Wu et al. (2018) states that SGD is stable at an iteration $t$ if the following inequality is satisfied:

$$(1 - \eta\lambda_H^1(t))^2 + s^2(t)\frac{\eta^2(n - S)}{S(n - 1)} \leq 1, \tag{2}$$

where $s(t) = \text{Var}[\text{H}_i(t)]$. Using Assumption 1 we get $\text{Var}[g_i(\psi)] = \text{Var}[\psi H_i] = \psi^2 s(t)^2$. Using this we can rewrite inequality (2) as:

$$(1 - \eta\lambda_H^1(t))^2 + \frac{\lambda_K^1(t)}{\psi(t)^2}\frac{\eta^2(n - S)}{S(n - 1)} \leq 1. \tag{3}$$

To calculate the spectral norm of the Hessian at iteration $t^*$ at which training becomes unstable we equate the two sides of inequality (3) and solve for $\lambda_H^1(t^*)$, which gives

$$\lambda_H^1(t^*) = \frac{2 - \frac{\alpha}{\psi(t^*)}\frac{(n-S)}{(n-1)}\frac{\eta}{S}}{\eta}, \tag{4}$$

where $\alpha$ denotes the proportionality constant from Assumption 2. Note that if $n = S$ the right hand side degenerates to $\frac{2}{\eta}$, which completes the proof for $n = S$.

To prove the general case, let denote by $\lambda_H^1(t_1^*)$ and $\lambda_H^1(t_2^*)$ the value of $\lambda_H^1$ at which training becomes unstable along $e_H^1$ for $\eta_1$ and $\eta_2$, respectively. Similarly, let us denote by $\psi(t_1^*)$ and $\psi(t_2^*)$ the corresponding values of $\psi$.

Let us assume by contradiction that $\lambda_H^1(t_1^*) > \lambda_H^1(t_2^*)$. A necessary condition for this inequality to hold is that $\psi(t_1^*) > \psi(t_2^*)$. However, Assumption 4 implies that prior to reaching the break-even point $\psi(t)$ decreases monotonically with increasing $\lambda_H^1(t_1^*)$, which corresponds to reducing distance to the minimum along $e_H^1$, which contradicts $\lambda_H^1(t_1^*) > \lambda_H^1(t_2^*)$. Repeating the same argument for two trajectories optimized using two different batch sizes completes the proof.

$\square$

It is also straightforward to extend the argument to the case when training is initialized at an unstable region along $e_H^1(0)$, which we formalize as follows.

**Theorem 2.** *If training is unstable along $e_H^1(t)$ at $t = 0$, then $\lambda_H^1(t)$ and $\lambda_K^1(t)$ at which SGD becomes for the first time stable along $e_H^1(t)$ are smaller for a larger $\eta$ or a smaller $S$.*

*Proof.* We will use a similar argument as in the proof of Th. 1. Let us consider two optimization trajectories corresponding to using two different learning rates $\eta_1$ and $\eta_2$ ($\eta_1 > \eta_2$).

Let $\lambda_H^1(t_1^*)$ and $\lambda_H^1(t_2^*)$ denote the spectral norm of $\mathbf{H}$ at the iteration at which training is for the first time stable along $e_H^1$ for $\eta_1$ and $\eta_2$, respectively. Following the steps in the proof of Th. 1 we get that

$$\lambda_H^1(t^*) = \frac{2 - \frac{\alpha}{\psi(t^*)} \frac{(n-S)}{(n-1)} \frac{\eta}{S}}{\eta}. \tag{5}$$

Using the same notation as in the proof of Th. 1, let us assume by contradiction that $\lambda_H^1(t_1^*) > \lambda_H^1(t_2^*)$. Assumption 3 implies that $\psi(t)$ increases with decreasing $\lambda_H^1(t)$. Hence, if $\lambda_H^1(t_2^*)$ is smaller, it means $\psi(t_2^*)$ increased to a larger value, i.e. $\psi(t_2^*) > \psi(t_1^*)$. However, if $\frac{(n-S)}{(n-1)} \approx 1$, a necessary condition for $\lambda_H^1(t_1^*) > \lambda_H^1(t_2^*)$ inequality to hold is that $\psi(t_1^*) > \psi(t_2^*)$, which leads to a contradiction. Repeating the same argument for batch size completes the proof.

$\square$

## C  APPROXIMATING THE EIGENSPACE OF $\mathbf{K}$ AND $\mathbf{H}$

Using a small subset of the training set suffices to approximate well the largest eigenvalues of the Hessian on the CIFAR-10 dataset (Alain et al., 2019).[3] Following Alain et al. (2019) we use approximately the same $5\%$ fraction of the dataset in CIFAR-10 experiments, and the SCIPY Python package. We use the same setting on the other datasets as well.

Analysing the eigenspace of $\mathbf{K}$ is less common in deep learning. For the purposes of this paper, we are primarily interested in estimating $\lambda_K^*$ and $\lambda_K$. We also estimate $\mathrm{Tr}(\mathbf{K})$. It is worth noting that $\mathrm{Tr}(\mathbf{K})$ is related to the variance in gradients as $\frac{1}{D}\mathrm{Tr}(\mathbf{K}) = \frac{1}{D}\frac{1}{N}\sum_{i=1}^{N}||g - g_i||^2$, where $g$ is the full-batch gradient, $D$ is the number of parameters and $N$ is the number of datapoints.

The naive computation of the eigenspace of $\mathbf{K}$ is infeasible for realistically large deep networks due to the quadratic cost in the number of parameters. Instead, we compute $\mathbf{K}$ using mini-batches. To avoid storing a $D \times D$ matrix in memory, we first sample $L$ mini-batch gradient of size $M$ and compute the corresponding Gram matrix $\mathbf{K}^M$ that has entries $\mathbf{K}_{ij}^M = \frac{1}{L}\langle g_i - g, g_j - g\rangle$, where $g$ is the full-batch gradient, which we estimate based on the $L$ mini-batches. To compute the eigenspace of $\mathbf{K}^M$ we use SVD routine from the NumPy package. We take the $(L-1)^{th}$ smallest eigenvalue as the smallest non-zero eigenvalue of $\mathbf{K}^M$ (covariance matrix computed using $L$ observations has by definition $L - 1$ non-zero eigenvalues).

Papyan (2019); Fort & Ganguli (2019) show that the top eigenvalues of $\mathbf{H}$ emerge due to clustering of gradients of the logits. See also Fort & Ganguli (2019). Based on the proximity of the largest eigenvalues of $\mathbf{H}$ and $\mathbf{K}$, this observation suggests that the top eigenvalue of $\mathbf{K}$ might be similar to that of $\mathbf{K}^M$. To investigate this, we run the following experiment on the CIFAR-10 dataset using

---

[3]This might be due to the *hierarchical* structure of the Hessian (Papyan, 2019; Fort & Ganguli, 2019). In particular, they show that the largest eigenvalues of the Hessian are connected to the class structure in the data.

SimpleCNN. We estimate $\lambda_K^1$ using $M = 1$ and $M = 128$, in both cases using the whole training set. We subsample the dataset to $10\%$ to speed up the computation. Fig. 9 shows a strong correlation between $\lambda_K^1$ computed with $M = 1$ and with $M = 128$, for three different learning rates.

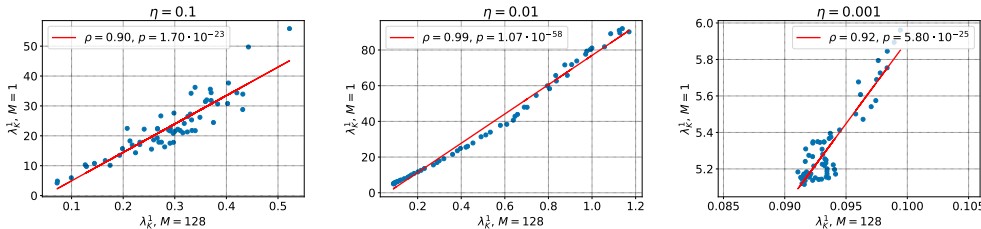

Figure 9: Pearson correlation between $\lambda_K^1$ calculated using either $M = 128$ or $M = 1$ for three different values of $\eta = 0.1, 0.01$ and $0.001$.

In all experiments we use $L = 25$, and for the IMDB dataset we increase the number of mini-batch gradients to $L = 200$ to reduce noise (same conclusion hold for $L = 25$). For instance, on the CIFAR-10 dataset this amounts to using approximately $5\%$ of the training set. In experiments that vary the batch size we use $M = 128$. Otherwise, we use the value $M$ as the batch size used to train the model.

## D   EXPERIMENTAL DETAILS FOR SEC. 4.2

In this section we describe all the details for experiments in Sec. 4.2.

**ResNet-32 on CIFAR-10.**   ResNet-32 (He et al., 2015) is trained for 200 epochs with a batch size equal to 128 on the CIFAR-10 dataset. Standard data augmentation and preprocessing is applied. Following He et al. (2015), we regularize the model using weight decay 0.0001. We apply weight decay to all convolutional kernels. When varying the batch size, we use learning rate of 0.05. When varying the learning rate, we use batch size of 128.

**SimpleCNN on CIFAR-10.**   SimpleCNN is a simple convolutional network with four convolutional layers based on Keras examples repository (Chollet et al., 2015). The architecture is as follows. The first two convolutional layers have 32 filters, and the last two convolutional layers have 64 filters. After each pair of convolutional layers, we include a max pooling layer with a window size of 2. After each layer we include a ReLU nonlinearity. The output of the final convolutional layer is processed by a densely connected layer with 128 units and ReLU nonlinearity. When varying the batch size, we use a learning rate of 0.05. When varying the learning rate, we use a batch size of 128.

**BERT on MNLI.**   The model used in this experiment is the BERT-base from Devlin et al. (2018), pretrained on multilingual data[4]. The model is trained on the MultiNLI dataset (Williams et al., 2018) with the maximum sentence length equal to 40. The network is trained for 20 epochs using a batch size of 32. Experiments are repeated with three different seeds that control initialization and data shuffling.

**MLP on FashionMNIST.**   This experiment is using a multi-layer perceptron with two hidden layers of size 300 and 100, both with ReLU activations. The data is normalized to the $[0, 1]$ range. The network is trained with a batch size of 64 for 200 epochs.

**LSTM on IMDB.**   The network used in this experiment consists of an embedding layer followed by an LSTM with 100 hidden units. We use vocabulary size of 20000 words and the maximum

---

[4]The model weights used can be found at `https://tfhub.dev/google/bert_multi_cased_L-12_H-768_A-12/1`.

length of the sequence equal to 80. The model is trained for 100 epochs. When varying the learning rate, we use batch size of 128. When varying the batch size, we use learning rate of 1.0. Experiments are repeated with two different seeds that control initialization and data shuffling.

**DenseNet on ImageNet.**   The network used is the DenseNet-121 from Huang et al. (2016). The dataset used is the ILSVRC 2012 (Russakovsky et al., 2015). The images are centered and normalized. No data augmentation is used. Due to large computational cost, the network is trained only for 10 epochs using a batch size of 32.

# E   ADDITIONAL EXPERIMENTS FOR SEC. 4.2.

In this section we include additional data for experiments in Sec. 4.2, as well as include experiments using MLP trained on the Fashion MNIST dataset.

**SimpleCNN on CIFAR-10.**   In Fig. 12 and Fig. 13 we report accuracy on the training set and the validation set, $\lambda_H^1$, and $\mathrm{Tr}(\mathbf{K})$ for all experiments with SimpleCNN model on the CIFAR-10 dataset .

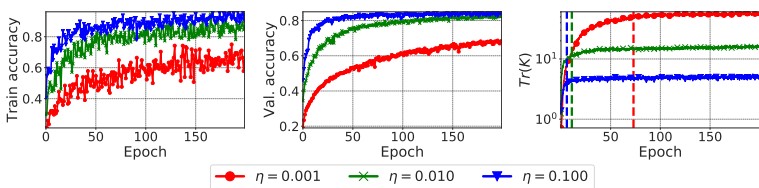

Figure 10: Additional metrics for the experiments using SimpleCNN on the CIFAR-10 dataset with different learning rates. From left to right: training accuracy, validation accuracy, $\mathrm{Tr}(\mathbf{K})$.

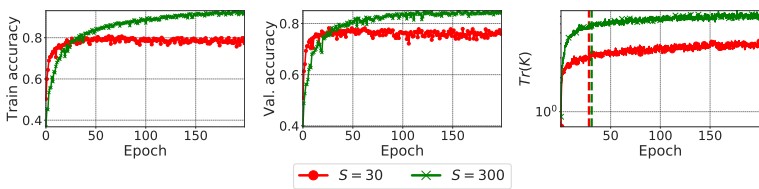

Figure 11: Additional metrics for the experiments using SimpleCNN on the CIFAR-10 dataset with different batch sizes. From left to right: training accuracy, validation accuracy, and $\mathrm{Tr}(\mathbf{K})$.

**ResNet-32 on CIFAR-10.**   In Fig. 12 and Fig. 13 we report accuracy on the training set and the validation set, $\lambda_H^1$, and $\mathrm{Tr}(\mathbf{K})$ for all experiments with ResNet-32 on the CIFAR-10 dataset.

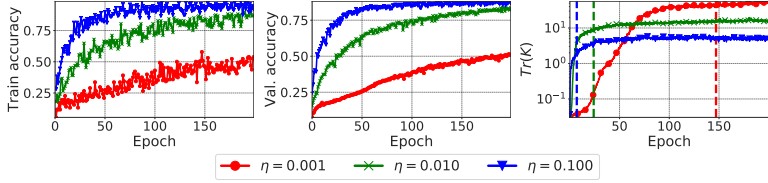

Figure 12: Additional figures for the experiments using ResNet-32 on the CIFAR-10 dataset with different learning rates. From left to right: the evolution of accuracy, validation accuracy, $\mathrm{Tr}(\mathbf{K})$.

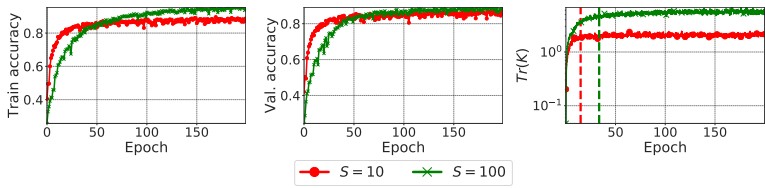

Figure 13: Additional figures for the experiments using ResNet-32 on the CIFAR-10 dataset with different batch sizes. From left to right: training accuracy, validation accuracy, $\mathrm{Tr}(\mathbf{K})$.

**LSTM on IMDB.** In Fig. 14 and Fig. 15 we report accuracy on the training set and the validation set, $\lambda_H^1$, and $\mathrm{Tr}(\mathbf{K})$ for all experiments with LSTM on the IMDB dataset.

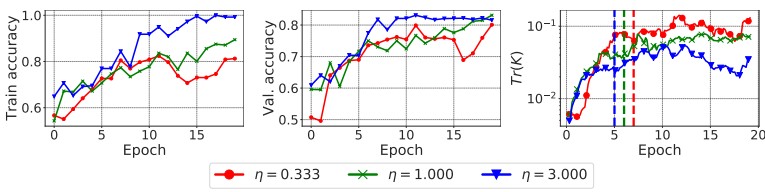

Figure 14: Additional figures for the experiments using LSTM on the IMDB dataset with different learning rates. From left to right: the evolution of accuracy, validation accuracy, and $\mathrm{Tr}(\mathbf{K})$.

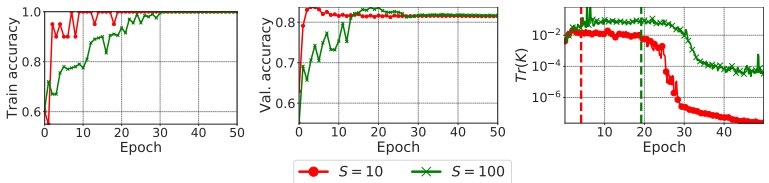

Figure 15: Additional metrics for the experiments using LSTM on the IMDB dataset with different batch sizes. From left to right: training accuracy, validation accuracy, $\lambda_H^1$ and $\mathrm{Tr}(\mathbf{K})$.

**BERT on MNLI.** In Fig. 16 we report accuracy on the training set and the validation set and $\mathrm{Tr}(\mathbf{K})$ for BERT model on the MNLI dataset.

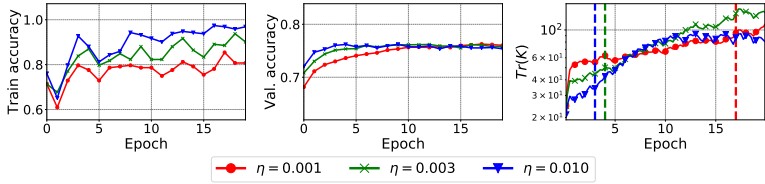

Figure 16: Additional metrics for the experiments using BERT on the MNLI dataset with different learning rates. From left to right: training accuracy, validation accuracy, and $\mathrm{Tr}(\mathbf{K})$.

**DenseNet on ImageNet.** In Fig. 17 we report accuracy on the training set and the validation set and $\mathrm{Tr}(\mathbf{K})$ for DenseNet on the ImageNet dataset.

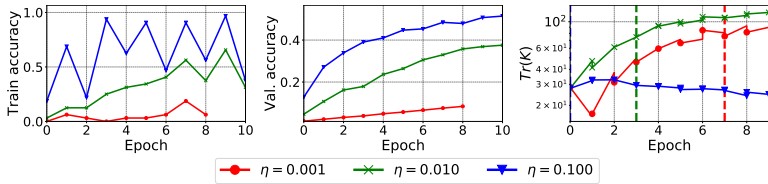

Figure 17: Additional metrics for the experiments using DenseNet on the ImageNet dataset with different learning rates. From left to right: training accuracy, validation accuracy, and $\mathrm{Tr}(\mathbf{K})$.

**MLP on FashionMNIST.** In Fig. 18 and Fig. 19 we report results for MLP model on the FashionMNIST dataset. We observe that all conclusions carry over to this setting.

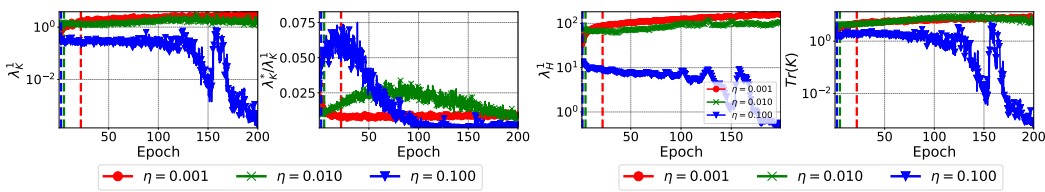

Figure 18: Results of experiment using the MLP model on the FashionMNIST dataset for different learning rates. From left to right: $\lambda_K^1$, $\lambda_K^*/\lambda_K^1$, $\lambda_H^1$ and $\mathrm{Tr}(\mathbf{K})$.

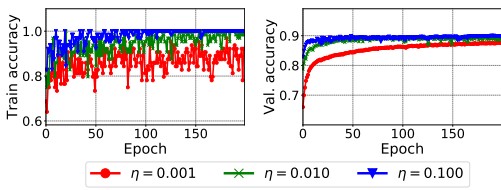

Figure 19: Training accuracy and validation accuracy for experiment in Fig. 18.

# F   ADDITIONAL EXPERIMENTS FOR SGD WITH MOMENTUM

Here, we run the same experiment as in Sec. 4.2 using SimpleCNN on the CIFAR-10 dataset. Instead of varying the learning rate, we test different values of the momentum $\beta$ parameter in the range of $0.1$, $0.5$ and $0.9$. We can observe that Conjecture 1 and Conjecture 2 generalize to momentum in the sense that using a higher momentum has an analogous effect to using a higher learning rate, or using a smaller batch size in SGD. We report the results in Fig. 20 and Fig. 21.

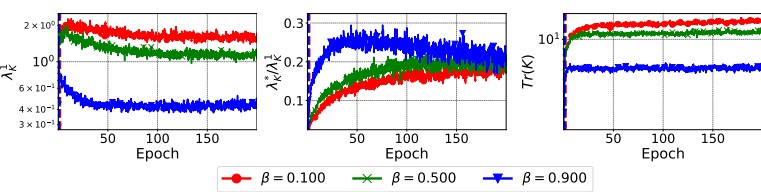

Figure 20: The variance reduction and the pre-conditioning effect for SimpleCNN trained using SGD with momentum. From left to right: $\lambda_K^1$, $\lambda_K^*/\lambda_K^1$ and $\mathrm{Tr}(\mathbf{K})$.

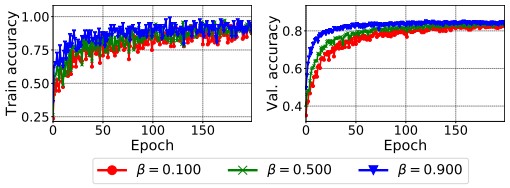

Figure 21: Training accuracy and validation accuracy for the experiment in Fig. 20.

Next, to study whether our Conjectures are also valid for SGD with momentum, but held constant, we run the same experiment as in Sec. 4.2 using SimpleCNN on the CIFAR-10 dataset. For all runs, we set momentum to $0.9$. Learning rate $0.1$ diverged training, so we include only $0.01$ and $0.001$. We can observe that both Conjectures generalize to this setting. We report the results in Fig. 22 and Fig. 23.

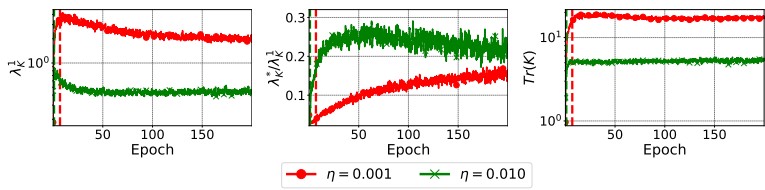

Figure 22: The variance reduction and the pre-conditioning effect for SimpleCNN trained using SGD with momentum. From left to right: $\lambda_K^1$, $\lambda_K^*/\lambda_K^1$ and $\mathrm{Tr}(\mathbf{K})$.

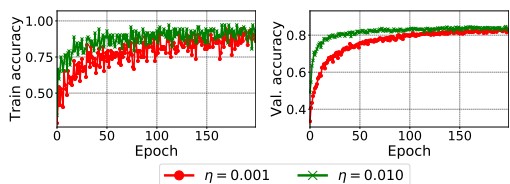

Figure 23: Training accuracy and validation accuracy for the experiment in Fig. 22.

## G  ADDITIONAL EXPERIMENTS FOR SGD WITH LEARNING RATE DECAY

To understand the effect of learning rate decay, we run the same experiment as in Sec. 4.2 using SimpleCNN on the CIFAR-10 dataset. Additionally, we divide the learning rate by the factor of $10$ after 100th epoch. We can observe that Conjecture 1 and Conjecture 2 generalize to scenario with learning rate schedule in the sense that changing learning rate doesn't change the relative ordering of the maximum $\lambda_K^1$ and $\lambda_K^*/\lambda_K^1$. We report the results in Fig. 24 and Fig. 25.

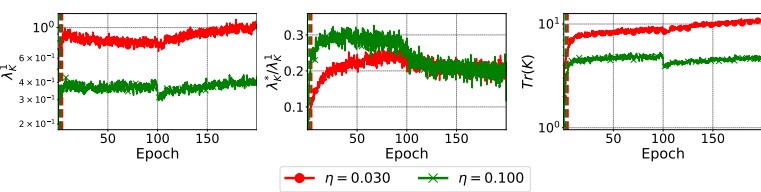

Figure 24: The variance reduction and the pre-conditioning effect for SimpleCNN trained using SGD with learning rate schedule. From left to right: $\lambda_K^1$, $\lambda_K^*/\lambda_K^1$ and $\mathrm{Tr}(\mathbf{K})$.

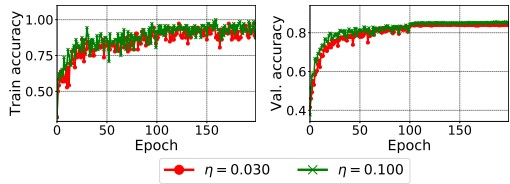

Figure 25: Training accuracy and validation accuracy for the experiment in Fig. 24.

# H  ADDITIONAL EXPERIMENTS FOR SIMPLECNN-BN

To further explore the connection between our conjectures and the effects of batch-normalization layers on the conditioning of the loss surface, we repeat here experiments from Sec. 4.3, but varying the batch size. Fig. 26 summarizes the results.

On the whole, conclusions from Sec. 4.3 carry over to this setting in the sense that decreasing the batch size has a similar effect on the studied metrics as increasing the learning rate. One exception is the experiment using the smallest batch size of 10. In this case, the maximum values of $\frac{\|g\|}{\|g_5\|}$ and $\lambda_K^*/\lambda_K^1$ are smaller than in the experiments using larger batch sizes.

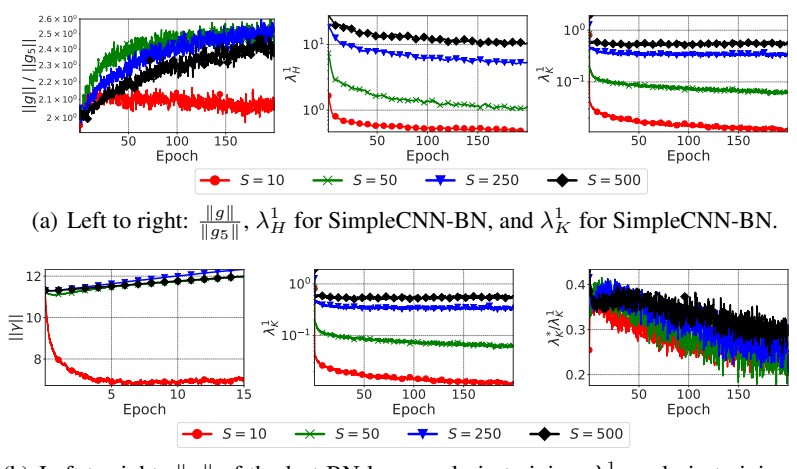

(a) Left to right: $\frac{\|g\|}{\|g_5\|}$, $\lambda_H^1$ for SimpleCNN-BN, and $\lambda_K^1$ for SimpleCNN-BN.

(b) Left to right: $\|\gamma\|$ of the last BN layer early in training, $\lambda_K^1$ early in training, $\lambda_K^*/\lambda_K^1$ for SimpleCNN-BN.

Figure 26: Evolution of various metrics that quantify conditioning of the loss surface for SimpleCNN with with batch normalization layers (SimpleCNN-BN), for different batch sizes.

