# OpenReview forum: "The Break-Even Point on Optimization Trajectories of Deep Neural Networks"
_ICLR.cc/2020/Conference — Accept (Spotlight)_

### Official Review · AnonReviewer1 · 2019-10-21
**Official Blind Review #1**

**Rating:** 6

**Review:**

This work analyzes the optimization of deep neural networks from the point of view of the different learning trajectories obtained during different learning settings as brought about by different hyperparameters in optimization. Specifically, the authors consider how the batch size (S) and the step-size (eta) hyper-parameters modify the learning trajectory. The authors conduct their analysis for networks with and without BatchNorm.

To get a quantitative understanding of the different trajectories of the optimization landscape the authors monitor and analyze the
A.  Hessian of the network with respect to the parameters
B. The covariance matrix of the per-sample gradients
The majority of the work focuses on monitoring A and B for various hyper-parameter regimes.
The authors discover a nice and clear trend, the leading eigenvalues of H (the Hessian) and K (per sample gradient covariance) are
1. Heavily correlated to each other (Figure.2 left)
2. Positively correlated with batch size (S) (Figures 3-5)
3. Negatively correlated with step-size (eta) (Figures 3-5)

The 3 trends presented empirically all originate and qualitatively agree very well with equation (1) of the paper that describes stability regions in training.  Equation (1) directly prescribes the maximum allowed eigenvalue for the Hessian matrix H for a given a learning rate, and batch-size to ensure stability. For equation (1) to hold, the authors make  the assumption that H and K share a leading eigenvector direction (this assumption is partially corroborated on the CIFAR10 dataset). The empirical analysis presented in the paper analyzes training on the CIFAR10, ImageNet, IMDB, and the MNLI datasets and uses a variety of common network architectures to evaluate the empirical claims. In general, I am quite happy with the breadth and depth of the experimental results presented in the paper!

The paper makes a few observations in addition to trends 1-3, It argues that BatchNorm networks need a large learning rate to exhibit smaller dependence on the subspace of the leading eigenvectors of K,  and that the networks reach a breaking point early in training where the leading eigenvalues of H and K are chosen by the trajectory.

Overall the paper is well written and the numerous experimental results are quite impressive. Nonetheless I have a few problems that are discouraging me from fully accepting this paper, so I am currently borderline accepting this paper.

The following problems should be addressed:

i. The theoretical results in the paper are very similar to the results given in Wu et al. (as mentioned in the main body and Appendix A). Moreover, the modified proofs presented in the appendix are not fully explained and are difficult to follow so I do not find them rigorous. It also seems like some of the empirical analysis is also presented in Wu et al.

ii. As stated, Conjecture 2 is not empirically supported in the paper. Currently, Conjecture 2 proposes that for  training with smaller batches or larger learning rates, the reciprocal of the ''conditioning number'' ( least POSITIVE eigenvalue / leading eigenvalue)  reaches a larger maximum value during training. Instead the authors "estimate" the least positive eigenvalue by the trace of the matrix which is very much a different notion than the least positive eigenvalue and also displays a different behavior.
It seems maybe measuring the trace as an un-normalized quantity for the average eigenvalue is more suitable but that is different from the least positive eigenvalue. Perhaps of the same flavor would be a statement on the stable rank of the matrices in question.


iii. Figure 1 (left) is quite confusing. I hope the authors can help clarify the heatmap shown along with the trajectories. The caption mentions that the colors signify the value of the leading eigenvalue of K.  First, since this is an explicit quantity can the authors provide a value range for the heatmap? Second, and more importantly, how is this quantity, which depends on K, can be computed for all the points in the 2D grid of UMAP to form this heat map? Recall K is a quantity that is computed for a given parameterization value and fixed number of mini-batch examples.

iv. Lack of clarity and rigor for the eigenvalue approximation: In appendix B the authors discuss approximating the eigenvalues of K and H, I think it is okay to develop numerically efficient approximations to the quantities in question but the presentation is not very intuitive nor rigorous. In the first paragraph the authors mention that since H is ill conditioned, a small subset of the training examples gives a good approximation to the Hessian, can the authors elaborate on this statement?

Other more minor comments:

Even though the default step size and batches are mentioned in appendix D it could be nice to mention them in the figures that they are presented in.

Page 7 Fig 5b | The labels of the figures should be adjusted so that they do not overlap with the adjacent figures.

Minor typos/grammar:
page 1 caption of figure 1 | vertical line marks -> the vertical line marks
page 3 in definitions paragraph | let us denote loss -> let us denote the loss
page 3 3rd assumption | training trajectory -> the training trajectory
page 3, 3rd assumption | escape region-> escape a region
page 7, other experiments paragraph | depend -> depends,  possibly?
page 7, Figure 5 (b) Middle |  lambda_k to lambda_k^1

**Experience Assessment:**

I have read many papers in this area.

**Review Assessment: Checking Correctness Of Derivations And Theory:**

I carefully checked the derivations and theory.

**Review Assessment: Checking Correctness Of Experiments:**

I carefully checked the experiments.

**Review Assessment: Thoroughness In Paper Reading:**

I read the paper thoroughly.

---

> ### Author Response · Authors · 2019-11-09
> **Response 2/2**
>
> > iii. Figure 1 (left) is quite confusing. I hope the authors can help clarify the heatmap shown along with the trajectories. The caption mentions that the colors signify the value of the leading eigenvalue of K.  First, since this is an explicit quantity can the authors provide a value range for the heatmap?
>
> We agree and have revised Fig 1 to show the heatmap. To further improve clarity, we replaced the right-hand plot with plot of training accuracy, and updated the caption to clarify the main take-away from Figure 1.
>
> > Second, and more importantly, how is this quantity, which depends on K, can be computed for all the points in the 2D grid of UMAP to form this heat map? Recall K is a quantity that is computed for a given parameterization value and fixed number of mini-batch examples.
>
> We appologite for the omission. The background color corresponds to the spectral norm (or accuracy) of the closest iteration. We have clarified this in the paper.
>
> > Lack of clarity and rigor for the eigenvalue approximation: the presentation is not very intuitive nor rigorous. In the first paragraph the authors mention that since H is ill conditioned, a small subset of the training examples gives a good approximation to the Hessian, can the authors elaborate on this statement?
>
> Indeed it wasn’t clear. What we intended to say is that the top eigenvectors of H are disproportionately more stable (i.e. are shared between Hessians computed on smaller subset of the data) than the lower eigenvectors. This arises from the hierarchical structure of the Hessian, as shown in Papayan [3]. This is a likely explanation for why, as shown by Alain et al [4], a small subset of the data already approximates well the largest eigenvalues of the Hessian.
>
> We have revised the paper to clarify the above. Furthermore, to improve rigour, we extended the experimental results. Now we compare our approximation against the exact K.
>
> >Other more minor comments:
>
> Thank you again for the review. We have addressed all the remaining comments.
>
> References
>
> [1] Fort et al, “Emergent properties of the local geometry of neural loss landscapes”, https://arxiv.org/abs/1910.05929.
> [2] Zhu et al, “The Anisotropic Noise in Stochastic Gradient Descent: Its Behavior of Escaping from Sharp Minima and Regularization Effects”,  http://proceedings.mlr.press/v97/zhu19e/zhu19e.pdf
> [3] Papayan et al, “Measurements of Three-Level Hierarchical Structure in the Outliers in the Spectrum of Deepnet Hessians”, https://arxiv.org/abs/1901.08244
> [4] Alain et al, “Negative eigenvalues of the Hessian in deep neural networks”, https://arxiv.org/abs/1902.02366

---

> ### Author Response · Authors · 2019-11-09
> **Response 1/2**
>
> > In general, I am quite happy with the breadth and depth of the experimental results presented in the paper! Overall the paper is well written and the numerous experimental results are quite impressive. Nonetheless I have a few problems that are discouraging me from fully accepting this paper, so I am currently borderline accepting this paper.
>
> Thank you very much for the review! We include below our answers, and please let us know if you have any further comments or questions.
>
> > The theoretical results in the paper are very similar to the results given in Wu et al.
>
> In essence, we use their theorem, but with a very different goal in mind.
>
> Most importantly, our conjectures *are not* an immediate consequence of the work of Wu et al. This is most clearly visible in the case of Conjecture 2. We model and study the evolution of H and K along the training trajectory, while Wu et al. focus on the properties of the final minimum.
>
> Practically speaking, we focus on the early phase of training, while Wu et al. focus on the final phase of training. For example, Wu et al. study the escape process from the minimum, while we study the break-even point, which occurs early in training (e.g. on the CIFAR-10 dataset, around reaching 50% of training accuracy). We have clarified this in the paper.
>
> > Moreover, the modified proofs presented in the appendix are not fully explained and are difficult to follow so I do not find them rigorous
>
> We apologize for the lack of rigour in the proofs. We have edited them with clarity and rigour in mind. We have also added an assumption that distance to the minimizer along e_H^1 decreases (increases) with increasing (decreasing) \lambda_H^1.
>
> > It also seems like some of the empirical analysis is also presented in Wu et al.
>
> While Fig. 8 and Fig. 9 in Wu et al show the evolution of the Hessian, the key difference is that they analyse these metrics after “escaping” a minimum. Hence, Wu et al does not study the training trajectory, but rather they study the escape process (similarly to [2]).
>
> > ii. As stated, Conjecture 2 is not empirically supported in the paper.
>
> This is a very good point. We agree that Tr(K) or Tr(H) are more directly related to the average eigenvalue of K or H, and hence the empirical results did not directly test Conjecture 2. To directly test Conjecture 2, we have replotted all the Figures using the ratio of the (5th) smallest eigenvalue and the largest eigenvalue. Furthermore, for experiment in 4.1 we computed this for H and included the plot in the submission. We think that Conjecture 2 is empirically supported in the revised version. Please let us know if any other experimental data would be useful.

---

### Official Review · AnonReviewer2 · 2019-10-24
**Official Blind Review #2**

**Rating:** 8

**Review:**

The authors demonstrate that, during training, there is a point during the early phase of training that leads stochastic gradient descent (SGD) to a point where the covariance of the gradients (K) has a lower spectral norm (smaller first eigenvalue) and improved conditioning in K and the Hessian of the training loss (H).

The authors experiments seem to verify that learning rate and batch size do play a part in the spectral norm of K and the conditioning of K. My one issue is that, while effects on K produced by higher learning rates are supposed to be "good", the authors do not directly relate this back to model performance. From my years of experience training neural networks, I have seen many scenarios in which higher learning rates result in worse performance, even after reducing the learning rate. Can this be related back to the author's claims? Under what conditions does a higher learning rate lead to these effects on K and H and will it always lead to better model performance?


Other comments:
In definitions, it says that the eigenvalue of matrix A is \lambda_A^i, however, later in the 4th part of the assumptions, the spectral norm of H is referred to as \lambda_1^H. Is there a difference here? Typo?

The last part of the definitions where \Phi(\tau) is introduced should have a formal definition for \Phi(\tau) as \Phi is initially does not take any parameter \tau.

In section 4.1, "further growth of \lambda_K^1 K does not translate into an increase of \lambda_K^1" \lambda_K^1 is repeated. Typo?

** After author response **
Changing from weak accept to accept.

The authors have addressed my concerns about the paper.

**Experience Assessment:**

I do not know much about this area.

**Review Assessment: Checking Correctness Of Derivations And Theory:**

I did not assess the derivations or theory.

**Review Assessment: Checking Correctness Of Experiments:**

I assessed the sensibility of the experiments.

**Review Assessment: Thoroughness In Paper Reading:**

I made a quick assessment of this paper.

---

> ### Author Response · Authors · 2019-11-09
> **Response**
>
> Thank you very much for the positive review. We include below our answers, and please let us know if you have any further comments or questions.
>
> >From my years of experience training neural networks, I have seen many scenarios in which higher learning rates result in worse performance
>
> Indeed, increasing learning rate typically improves performance only up to some point, and after this point decreases performance. This phenomenon is analogously observed for explicit regularizers such as weight decay.
>
> >Can this be related back to the author's claims? Under what conditions does a higher learning rate lead to these effects on K and H and will it always lead to better model performance?
>
> This is a very good point. As shown in prior work [1,2], a too high learning rate can destabilize learning. There always exists a learning rate value such that training diverges after the first step. Hence, while high learning rate “improves” properties of H and K, after some point the destabilization effects (as discussed in [1,2]) start to dominate the positive effects on H and K. We have clarified this in the paper.
>
> Second, both effects on K and H are desirable from the optimization perspective, and achieving these effects is actually a common goal in the optimization literature. For instance [3] propose techniques to reduce variance (i.e. Euclidean distance between the mini-batch gradient and the full-batch gradient), and [4] uses the inverse of K to condition training. To expand on the second, optimizing in a region with more isotropic K means that the SGD step is more similar (in the Euclidean norm, for example) to a step that uses the inverse of K like in [4].
>
>
> >Other comments:
>
> Thank you for the comments. We have addressed all of these.
>
> References
>
> [1] Masters et al, “Revisiting Small Batch Training for Deep Neural Networks”, https://arxiv.org/pdf/1804.07612.pdf
> [2] Bjorck et al, “Understanding Batch Normalization”,  https://arxiv.org/abs/1806.02375
> [3] Johnson et al “Accelerating Stochastic Gradient Descent using Predictive Variance Reduction” https://papers.nips.cc/paper/4937-accelerating-stochastic-gradient-descent-using-predictive-variance-reduction.pdf
> [4] Roux et al “Topmoumoute Online Natural Gradient Algorithm” https://papers.nips.cc/paper/3234-topmoumoute-online-natural-gradient-algorithm

---

### Official Review · AnonReviewer3 · 2019-11-06
**Official Blind Review #3**

**Rating:** 6

**Review:**

This paper studies two objects that quantify the optimization trajectory: the Hessian of the training loss (H) that describes the curvature of the loss surface, and the covariance of gradients that quantifies noise induced by noisy estimate of the full-batch gradient.

The authors predict and demonstrate that learning rate and batch size determine H and K and demonstrate that both large learning rate and small batch size results in two effects on K and H along the trajectory: (1) variance reduction and (2) pre-conditioning. These effects are observed after the break-even point. The further verified these predictions on BN networks.

Comments:

Understanding the optimization trajectory is an important topic and this work is one step towards better understanding. There are many questions I hope the authors could clarify:

- the concept of break even point, which justify whether the minima is stable or not, is also presented in Wu et al, 2018, including the proof. In the introduction of equation 1, the break even point concept seems to be novel in the context of learning trajectory. What is the difference with Wu et al, 2018? Does this break even point appear only once during training?

- What is the practical usage for identifying the break-even point or stable/unstable of the optimizer? Does reaching the unstable phase of SGD earlier by large learning rate/small batch size mean better performance?

- It seems the author does not apply learning rate decay for all experiments. What is the connection with learning rate decay? As we often see the training loss and validation error decreases significantly during training in step based learning rate schedule, what is their corresponding change of \lambda_H^1 and \lambda_K^1?

- The authors did not mention whether momentum is used or not during training, which is commonly used in training neural nets. How would momentum affect the conjectures?

- Regarding to conjecture 1 which states that “larger” learning rate yields lower \lamda_H^1 and \lamda_K^1, is there a limit for the range of learning rate? If we use learning rate 10, the training may not even converge. The experiments only verifies three learning rate with a maximum value 0.1, which does not cover “large” learning rate.

- Also as small batch size naturally results in more iterations than large batch given the same number of epochs. Comparing small batch and large batch may need to take this into account. By reaching the break-even-point “early”, does it mean less number of epochs? I would like to see the comparison in terms of iterations rather than epochs.

- In section 4.1, the definition of \alpha* is not clear. It was noted in Figure 2 as the width of the loss surface and is defined to be the minimum step size along the adjacent iterate direction to have 20% loss increase. What is the unit of t here, is it a batch or epoch?
How exactly is \alpha calculated? Due to the scale invariance [1,2], this \alpha is not necessarily the true `width` of the loss surface as it could be influenced by the weight norm.
Why do different learning rates reach a similar \alpha? It would be better to mark the starting point of the trajectory and the ending point of trajectory.

- The illustration of Figure 1a is very unclearer. What is the x-axis and y-axis? Where is exactly the break-even point? It would be helpful to mark the starting and ending point of the trajectory. Also it would be helpful to have the values marked in the contours or make a separate value bar.


- The authors studied whether the conjectures hold for BN networks in section 4.3. They only verified learning rate but not batch size. Does the batch size part still hold? It is known that BN requires larger batch size to work well, which may contradict with the conjecture that smaller batch size works better.


Minor:
- I was confused by the Figure 3 where different hyperparameters has different values at epoch 0. Does epoch 0 means the first epoch? It would be clear to have the same starting point and make epoch 0 as the initialization point.

- The experiments of DenseNet on ImageNet seems incomplete as it is only trained for 10 epochs. What is the top-1 error on the validation set?

[1] https://arxiv.org/abs/1703.04933
[2] https://arxiv.org/abs/1712.09913


-----update after rebuttal----------

After reading the response, I think most of my questions are properly answered and corresponding changes are made in the revision. I  changed my score to weak accept. I would hope the authors could further improve the paper by well organizing the additional experiments and findings. Too much experiment variation settings make the paper not easy to follow. I hope the authors could make it clear about the practical benefits of identifying the existence of break even point with common experiments settings, e.g., nonzero momentum, weight decay and step based learning rate decay. In the new provided experiments about learning rate decay, I did not see the normal abrupt change of validation accuracy. More analysis about whether learning rate decay helps reaching break-even point would be great.

**Experience Assessment:**

I have published one or two papers in this area.

**Review Assessment: Checking Correctness Of Derivations And Theory:**

I assessed the sensibility of the derivations and theory.

**Review Assessment: Checking Correctness Of Experiments:**

I carefully checked the experiments.

**Review Assessment: Thoroughness In Paper Reading:**

I read the paper at least twice and used my best judgement in assessing the paper.

---

> ### Author Response · Authors · 2019-11-09
> **Response 3/3**
>
> >- The authors studied whether the conjectures hold for BN networks in section 4.3. They only verified learning rate but not batch size. Does the batch size part still hold? It is known that BN requires larger batch size to work well, which may contradict with the conjecture that smaller batch size works better.
>
> This is a very interesting point. In analogy to the detrimental effect of using a too high learning rate for any network, we agree that using a too small batch-size must be detrimental in batch-normalized networks.
> In batch-normalized networks batch size changes the loss landscape (by changing the forward pass for each example). This is not included in our theoretical model, which is the primary reason for why we didn’t include the experiment. We have clarified this in the paper, as well as that our Conjectures are applicable in the regime where training is stable.
>
> To investigate the effect of batch-size on K and H, we run the experiment in 4.3 for batch sizes of 10, 100, 250 and 500. We include detailed plots in https://bit.ly/34K2Qij.
>
> First, we observe that Con. 1 fully holds for all batch sizes we tried, both for lambda_H^1 and lambda_K^1. The norm of gamma in the last layer also decreases for a small batch size. Con. 2 seems to hold for batch-sizes larger than 10, batch size 10 seems to lead to very unstable learning dynamics (validation accuracy evolves . This is consistent with your remark, as well as the effect of a too high learning rate in normal networks. Indeed, looking at the training curves batch-size of 10 fails to optimize well the network.
>
> > I was confused by the Figure 3 where different hyperparameters has different values at epoch 0. Does epoch 0 means the first epoch? It would be clear to have the same starting point and make epoch 0 as the initialization point.
>
> We apologize for the confusion. The first tick on the x axis corresponds to the end of the 1st epoch. We will add the initialization point as well in the final version. In Figure 2 we included the initialization point; training starts at the same point in terms of \lambda_H^1 and \lambda_K^1.
>
> > The experiments of DenseNet on ImageNet seems incomplete as it is only trained for 10 epochs. What is the top-1 error on the validation set?
>
> The top-1 validation error has reached around 50% (at epoch 10) for LR=0.1.  However, reaching a low validation error was not necessary for this experiment to test Con.1 and Con.2. Computing our metrics for the whole training trajectory is unfortunately prohibitively expensive on such a large scale model and datasets. Thus we run the experiment for the minimal epochs that enabled us to test Con.1 and Con. 2.  After these 10 epochs it is visible that the peak of lambda_K (Con. 1) and of the conditioning (Con. 2) depends on the learning rate.
>
> To test better Con.1 and Con. 2 were also able also to add learning rate = 0.001, and added this to the paper. We also plotted the spectrum of K to better visualize Con. 2. We include plots in https://bit.ly/34K2Qij.
>
> —-
>
> We thank the reviewer again. We will also add the results from https://bit.ly/34K2Qij to the paper.
>
> References:
>
> [1] Johnson et al “Accelerating Stochastic Gradient Descent using Predictive Variance Reduction” https://papers.nips.cc/paper/4937-accelerating-stochastic-gradient-descent-using-predictive-variance-reduction.pdf
> [2] Roux et al “Topmoumoute Online Natural Gradient Algorithm” https://papers.nips.cc/paper/3234-topmoumoute-online-natural-gradient-algorithm
> [3] Erhan et al, Why does unsupervised pretraining help deep learning, http://www.jmlr.org/papers/volume11/erhan10a/erhan10a.pdf
> [4] Jastrzebski* Kenton* et al, “Three Factors Influencing Minima in SGD”, https://arxiv.org/abs/1711.04623
> [5] Keskar et al, “On Large-Batch Training for Deep Learning: Generalization Gap and Sharp Minima”, https://arxiv.org/abs/1609.04836
> [6] Chen et al, “Walk with SGD”, https://arxiv.org/abs/1802.08770

---

> ### Author Response · Authors · 2019-11-09
> **Response 2/3**
>
>
> > The authors did not mention whether momentum is used or not during training, which is commonly used in training neural nets. How would momentum affect the conjectures?
>
> Sorry for the omission. In the experiments we set momentum to 0. To answer your question, we run two additional experiments: with a constant momentum set to 0.9 and varying the momentum. These experiment show that (1) momentum acts in a similar way to learning rate or batch size in terms of the effects of H and K, and (2) a constant momentum > 0 doesn’t impact the conclusions. Plots are available at https://bit.ly/34K2Qij.
>
> > Regarding to conjecture 1 which states that “larger” learning rate yields lower \lamda_H^1 and \lamda_K^1, is there a limit for the range of learning rate? If we use learning rate 10, the training may not even converge. The experiments only verifies three learning rate with a maximum value 0.1, which does not cover “large” learning rate.
>
> Indeed, our assumptions implicitly enforce a constraint on the maximum learning rate; we require training to be able to find a stable region of the loss surface. In particular, if the learning rate is larger than the Lipschitz constant of the loss surface (assuming it exists), then training never finds a stable enough region, because it does not exist. Hence in such a case our conjectures cannot be satisfied.
>
> It is worth mentioning that learning rates larger than 0.1 in the CIFAR-10 experiments did make the optimization diverge, or were close to. We clarified this in the paper. We also include LR=1.0 in experiments with BN.
>
> > Also as small batch size naturally results in more iterations than large batch given the same number of epochs. Comparing small batch and large batch may need to take this into account. By reaching the break-even-point “early”, does it mean less number of epochs? I would like to see the comparison in terms of iterations rather than epochs.
>
> By reaching early the break-even point we meant reaching the break-even point at an iteration corresponding to a low value of the spectral norm of K (H).
>
> Importantly, in Con. 1 and Con. 2 we make predictions about the maximum values of the \lambda_H, \lambda_K, \lambda_H^*/\lambda_H^1, and \lambda_K^*/\lambda_K. These quantities are not affected by the scaling of the x-axis. We also include in https://bit.ly/34K2Qij version of Fig. 3 for SimpleCNN with changed horizontal axis. Please let us know if you would like to see any other plots in this format.
>
> > In section 4.1, the definition of \alpha* is not clear. It was noted in Figure 2 as the width of the loss surface and is defined to be the minimum step size along the adjacent iterate direction to have 20% loss increase. What is the unit of t here, is it a batch or epoch?
>
> We agree that Figure 2 is not very clear. Our main goal was to show that, as predicted by our theoretical model, optimization reaches a point in training characterized by large “instability”, which we quantified heuristically as the point where “it is easy to increase loss by just scaling up the step length”.
>
> To answer your questions: the time unit is one iteration. To compute \alpha^* we (1) take a step back, and (2) artificially take a larger step, (3) measure the loss change, (4) revert, i.e.reload the weights from before taking  the measurement. Regarding the scale effect: our metric is also scale invariant because as Th. 3 in Dinh et al. shows scaling up \theta1 (scaling down \theta2) by \alpha scales up (scales down) the gradient norm accordingly. Finally, different learning rates reaching a similar \alpha seems to corroborate our theoretical model. In our theoretical model along e_H^1 we do expect different learning rates to reach a similar \alpha at the break-even point.
>
> Based on your comment, we have replaced this experiment with a more direct one. Now we plotted the change in loss dL between two consecutive steps, and show that when \lambda_K^1 reaches the maximum value, dL starts to take negative values in some iterations.
>
> > The illustration of Figure 1a is very unclear. What is the x-axis and y-axis? Where is exactly the break-even point?
>
> We agree, and apologize for a rather confusing presentation of the experimental data. We have revised Figure 1 to include accuracies, contours, and the value bar. To answer your questions: x and y axis are UMAP coordinates (i.e. they are not semantically meaningful) - this plot was made exactly following the procedure used to plot Fig. 5 in Erhan et al [3]. The break-even point (for high LR) occurs at the iteration at which the \lambda_K^1 is at its maximum (we also extended Section 4.1 to better visualize reaching break-even point using other metrics). We hope this is now clearer in the new version of the Figure.

---

> ### Author Response · Authors · 2019-11-09
> **Response 1/3**
>
> Thank you very much for the review! Below are our responses and the results of the additional experiments can be accessed at https://bit.ly/34K2Qij. Please let us know if you have any further comments or questions.
>
> (We are sorry the rebuttal is rather long. The experiments in https://bit.ly/34K2Qij are described more concisely, and we include a summary of key changes in the global comment)
>
> >The concept of break even point, which justify whether the minima is stable or not, is also presented in Wu et al, 2018, including the proof. In the introduction of equation 1, the break even point concept seems to be novel in the context of learning trajectory. What is the difference with Wu et al, 2018?
>
> As you mentioned, the main difference between our work and Wu et al’s is that we focus on the training trajectory, rather than on the properties of the final minimum as Wu et al or [4, 5].
>
> Most importantly, our conjectures *are not* an immediate consequence of Wu et al. This is most clearly visible in Conjecture 2. We model and study the evolution of H and K along the training trajectory, while Wu et al focus on the properties of the final minimum.
> From the practical point of view, the effects that the two conjectures describe occur early in training. On CIFAR-10 that occurs when the model achieves around 50% training accuracy, as shown in (revised, in line with your comments) Figure 1, and influence most of the training trajectory.
>
> We updated the related work section to clarify these differences.
>
> >Does this break even point appear only once during training?
>
> In the theoretical model, we defined the break-even as the first time training becomes unstable (stable). We have clarified this in the paper. However, the dynamic we identified could repeat itself later in training, and in this sense there could be in principle multiple break-even points during training. We observe some signs of this happening in the learning rate decay experiment (see our answer later).
>
> > What is the practical usage for identifying the break-even point or stable/unstable of the optimizer?
>
> We did not investigate benefits of detecting that a break-even point has been reached. We demonstrated that reaching it “early” (i.e. at an iteration where \lambda_K^1 and \lambda_H^1 has not yet reached a large value, see also the answer to the next question) is beneficial for the optimization performance by reducing variance and improving conditioning (Con.1 and Con. 2).
>
> However, it seems that after reaching the break-even point, the training loss curvature is more predictable from iteration to iteration (due to the more stable evolution of \lambda_K^1 and \lambda_H^1). In principle, this information could be leveraged in designing second order optimizers.
>
> > Does reaching the unstable phase of SGD earlier by large learning rate/small batch size mean better performance?
>
> Reaching earlier the break-even point, i.e. at an iteration corresponding to a lower spectral norm of K (H) (see also our answer to the question on comparing at the same number of iterations) results in (1) variance reduction, and (2) better conditioning.
>
> Both effects are desirable from the optimization perspective, and achieving these effects is actually a common goal in the optimization literature. For instance [1] propose techniques to reduce variance (i.e. Euclidean distance between the mini-batch gradient and the full-batch gradient), and [2] uses the inverse of K to condition training. To expand on the second, optimizing in a region with more isotropic K means that the SGD step is more similar (in the Euclidean norm, for example) to a step that uses the inverse of K like in [2].
>
> >  It seems the author does not apply learning rate decay for all experiments. What is the connection with learning rate decay? As we often see the training loss and validation error decreases significantly during training in step based learning rate schedule, what is their corresponding change of \lambda_H^1 and \lambda_K^1?
>
> We agree that training loss and validation error tend to change rapidly after changing the learning rate. As a preliminary study for the effect of learning rate decay, we run the SimpleCNN model from Sec. 4.2 with base learning rate in 0.1 and 0.03, and a step-based learning rate schedule.
>
> In this case, dropping learning rate reduces \lambda_H^1 and \lambda_K^1. However, after a while the two quantities can increase, consistently with the experiment in Chen et al [4]. Importantly, the relative ordering of the magnitudes of \lambda_K and \lambda_H predicted in Con. 1 still holds. In this sense it seems that Con.1 (and Con. 2) can be extended to step-based learning rate schedules.
>
> It might be also worth mentioning that the common practice of warming up the learning rate is consistent with the conclusion from our Conjectures that it is beneficial to reach early the highest possible learning rate.
>
> We have include detailed plots in https://bit.ly/34K2Qij.

---

### Author Response · Authors · 2019-09-27
**Clarification**

We forgot to mention that ResNet-32 in Section 4.2 does not use batch normalization. In 4.3 we extend the analysis to batch normalized networks.

---

### Author Response · Authors · 2019-11-13
**Summary of key changes**

We thank again all reviewers for their time and comments. We include below a summary of the key changes:

In response to Reviewer 1 and Reviewer 3:
* We improved Fig. 1 to make it more readable
* We clarified that the main difference between us and Wu et al is that we apply a related methodology to the analysis of the whole training trajectory

In response to Reviewer 2 and Reviewer 3:
* We clarified why the predicted effects on H and K are desirable from the optimization perspective

In response to Reviewer 3:
* We improved Sec. 3 (mainly proofs in the Appendix) to be more rigorous and clearer
* We directly test and confirm Con. 2

We also run additional experiments and report them in https://bit.ly/34K2Qij.

---

### Decision · Program_Chairs · 2019-12-19

**Decision:**

Accept (Spotlight)

**Comment:**

This is an interesting study analyzing learning trajectories and their dependence on hyperparameters, important for better understanding of learning in deep neural networks.  All reviewers agree that the paper has a useful message to the ICLR community, and appreciate changes made by the authors in response to the initial reviews.